# Assessing the impact of multiple altimeter missions and Argo in a global eddy permitting data assimilation system

S. Verrier[1,2], P.Y. Le Traon[1,2], E. Remy[1]

[1]Mercator Ocean, Ramonville St Agne, 31520, France

[2]Ifremer, Plouzané, 29280, France

*Correspondence to*: S. Verrier (simon.verrier@mercator-ocean.fr)

**Abstract:** A series of Observing System Simulation Experiments (OSSEs) is carried out with a global data assimilation system at 1/4° resolution using simulated data derived from a 1/12° resolution free run simulation. The objective is to quantify how well multiple altimeter missions and Argo profiling floats can constrain the global ocean analysis and 7 day forecast  at 1/4° resolution but also to better understand the sensitivity of results to data assimilation techniques used in Mercator Ocean operational systems. Impact of multiple altimeter data is clearly evidenced even at a 1/4° resolution. 7 day forecasts of sea level and ocean currents are significantly improved when moving from one altimeter to two altimeters not only on the sea level but also on the 3D thermohaline structure and currents.  In high eddy energy regions, sea level and surface current 7 day forecast errors when assimilating one altimeter data set are respectively 20% and 45% of the error of the simulation without assimilation.  7 day forecasts of sea level and ocean currents continue to be improved when moving from one altimeter to two altimeters with a relative error reduction of almost 30%. The addition of a third altimeter still improves the 7 day forecasts even at this medium 1/4° resolution and brings an additional relative error reduction of about 10%.  The error level of the analysis with one altimeter is close to the 7 day forecast error level when two or three altimeter data sets are assimilated. Assimilating altimeter data also improves the representation of the 3D ocean fields. The addition of Argo has a major impact to improve temperature and demonstrates the essential role of Argo together with altimetry to constrain a global data assimilation system. Salinity fields are only marginally improved. Results derived from these OSSEs are consistent with those derived from experiments with real data (observing system evaluations/OSEs) but they allow a more detailed characterization of errors on analyses and 7 day forecasts. Both OSEs and OSSEs should be systematically used and intercompared to test data assimilation systems and quantify the impact of existing observing systems.

## 1. Introduction

Observing System Simulation Experiments (OSSEs) are powerful tools to evaluate the impact, relative merits and complementarities of the different components of the global ocean observing system. They allow an assessment of existing elements of the global ocean observing system and are essential to evaluate revised or new design (e.g. evolution of sampling characteristics, addition of a new observing system component). OSSEs rely on models that realistically represent the space-time variability of the essential ocean variables to be monitored, and data assimilation to optimally merge in-situ, satellite observations and models. OSSEs typically use two different models. One model is used to perform a "truth" or "nature" run - and it is treated as if it is the real ocean. The "nature" run is sampled in a manner that mimics either an existing or future observing system - yielding synthetic observations. The synthetic observations are assimilated into the

second model (assimilated run) and the model performance is evaluated by comparing it against the nature run. This in turn quantifies the impact of observations. OSSEs are also important tools to test the capability of global data assimilation systems to effectively merge different types of observations with models to produce improved ocean analyses and forecasts. OSSEs are complementary to OSEs (Observing System Evaluations). OSEs analyse the impact of real data for ocean analysis and forecasting generally by comparing a run assimilating all available data with a run assimilating all the data except for the data type to be investigated. OSSEs allow, however, a more comprehensive assessment of errors on analyses and forecasts at all depths and for all parameters through the comparison with the nature run (the "truth"). On the other hand, results for existing observing systems must be consistent with those derived from OSSEs. This issue of calibration of OSSEs with respect to OSEs is actually an important element for a proper design of OSSEs (e.g. Halliwell et al., 2014). Choice of the nature run, assimilated run, data assimilation scheme and errors to apply to synthetic observations should be carefully analysed to avoid under or overestimations of forecast and analysis errors in OSSEs.

In this study, an assessment of the impact of multiple altimeters and Argo profiling floats is carried out with the Mercator Ocean global 1/4° data assimilation system via a series of OSSEs. The objective is to quantify the impact of assimilating several altimeters on analyses and forecasts and the complementarities between altimetry and Argo observations when they are both assimilated. A secondary objective is to test the capability of the Mercator Ocean data assimilation system to effectively use and merge multiple altimeters and Argo. Altimetry and Argo are the backbone observing system required for operational oceanography (e.g. Le Traon, 2013). They are systematically used today to constrain global and regional ocean analysis and forecasting systems. Multiple altimeter missions are required to constrain the mesoscale circulation (e.g. Le Traon et al., 2015) and Argo observations are required to constrain the temperature and salinity fields. OSEs carried out in the context of the GODAE OceanView international program (Bell et al., 2015) have demonstrated the impact of assimilating several altimeters and Argo (e.g. ; Lea et al., 2014; Oke et al., 2015; Turpin et al., 2015). They show, in particular, that the addition of the first altimeter has the largest impact but that there are quantitative improvements seen by the addition of a second and third altimeter. Argo is, on the other hand, mandatory to constrain temperature and salinity fields (e.g. Turpin et al., 2015). Analysing the impact of altimetry and Argo in a global data assimilation system through OSSEs has, to our knowledge, not been carried out at least in recent years. Such an analysis can provide, however, very useful and complementary results compared to these past OSEs by allowing a more detailed analysis of analysis and forecast errors.

The paper is organized as follows. Section 2 provides a description of the OSSE methodology and modelling and data assimilation system. Section 3 analyses the impact of assimilating one, two or three altimeters. Complementary role of Argo is discussed in section 4. Main conclusions and future prospects are given in section 5.

## 2 OSSE methodology

This section describes the methodologies used to perform the different OSSEs. The Mercator Ocean data assimilation system is first presented. The Nature Run and the Free Run used to initialise the Assimilated Run, the simulation of observations and the characteristics of OSSEs are then described.

### 2.1 The Mercator Data Assimilation System

Commonly called SAM2, the current protocol for data assimilation at Mercator Ocean (Lellouche et al., 2013) computes correction over a 7 day assimilation window and is based on a modified Kalman filter named SEEK (Singular Evolutive Ensemble Kalman filter) first introduced by Pham et al. (1998). Analysis is calculated at the middle of the assimilation window, i.e. the fourth day. The SEEK filter means, as explained by Brasseur and Verron (2006), that covariance error matrices are forced at a low rank ("Singular") and that it computes model error covariances propagation ("Evolutive") following the model dynamics.

The filter used in SAM2 is not evolutive as a SEEK. Indeed, instead of using EOFs to build its error covariance matrix that will be propagated onto the model along time steps, SAM2 takes a fixed base of smoothed model anomaly fields (349 in the following experiments). This approach allows the system to get a covariance matrix that is realistic with the climatological statistics of the ocean model at the time step and saving computation time as this matrix will not be propagated in the model unlike the SEEK. Anomalies, for the five control variables (Sea Level, U, V, T, S), are calculated from a 10 year free model and at each date they are equal to the difference between the free run and a running mean along time over it-self. At the date of an analysis, only anomalies within the past 30 days and future 30 days and from the different years are considered. The final number of anomalies that are kept for given analysis is equal to 349. This means that the anomaly basis changes at each analysis date and follows the global model climatology. These anomalies are selected accordingly to the season of the assimilation cycle to get a basis evolving consistently with the model climatology. Our filter is not evolutive as the model error covariance is not propagated by the dynamical model. The model correction is calculated as a linear combination of the selected anomalies.. Then, this correction is injected linearly over the seven days using the *Incremental Analysis Update* (IAU, Bloom at al., 1996). As explained in Lellouche et al. (2013), when in situ measurements are assimilated, a bias correction based on a 3D-Var approach is used to correct large scale and slow evolving errors in T, S and thus dynamical height. Bias correction uses a collection of temperature and salinity innovations from the last three months and creates a correction to be added in the model prognostic equations. Here we kept the setup of the assimilation scheme as it is in the operational system and described in Lellouche et al. 2013 except for the next points : the representativity errors that we did not take into account, the assimilation of the full SSH signal and not only the SLA and the uniform observing error covariance matrix (3 cm in RMS).

## 2.2. The Nature Run and the initialisation of the Assimilated Run

In this study, both the "nature" run (NR) and the assimilated run (AR) are based on the NEMO model (Nucleus for European Modelling of the Ocean, Madec et al. 1998) with a global coverage and 50 vertical levels with 22 levels within the upper 100m and with 1m resolution for the first level up to 450 m for the last one (at the bottom). The system uses the OPA (*Océan Parallélisé*) model coupled with the LIM2 ice model (Fichefet and Morales Maqueda, 1997). The difference between the two configurations is that the NR uses a 1/12° tripolar grid (ORCA12) and the second one a 1/4° tripolar grid (ORCA025). Both models are forced using the CORE (Coordinated Ocean-Ice Reference Experiment) bulk formulation (Large and Yeager, 2009). The 1/12° free model is chosen for NR because it is a good estimation of the true ocean variability. The 1/12° NR was chosen for its capacity to better represent mesoscale variability (50-500km) in the ocean compared to a 1/4° resolution simulation (Hulburt et al. 2009). Assimilating data from a higher resolution model into the 1/4° configuration is a way to determine how these structures, underestimated in a free 1/4° model, can be forced to be closer to reality (NR).

The OSSEs were started from January 7, 2009 over an almost one-year time period. Two different initial conditions (i.e. January 7, 2009) for the NR and for the AR are required so that we can quantify the impact of assimilating pseudo observations of the NR in the AR. This was achieved by running the two free run NEMO configurations initialized from climatology but at different times. The NR simulation was started in 2003 and forced with ECMWF (European Centre of Medium Weather Forecasting) operational 3-h atmospheric data and the AR was initialized from a 1/4° free run started from 1989 and forced by ECMWF ERA-Interim 3h atmospheric data. The OSSEs are all forced with the ECMWF operational 3-h data. Note that as AR and NR are both forced by ECMWF operational data, our OSSEs do not address the impact of atmospheric forcing errors.

### 2.3 Simulated observations

To assess the impact of the number of altimeter data, three satellites have been considered: Jason-1, Jason-2 and Envisat (Fig. 1a). Jason-1 and Jason-2 have a 10-day repeat cycle and Envisat a 35-day repeat cycle. Jason-1 was in its interleaved

orbit with its ground tracks just in between Jason-2 tracks and a time shift of 5 days. This orbit was chosen to optimize mesoscale variability sampling by Jason-1 and Jason-2. The OSSEs were carried out over the year 2009. Jason-1, Jason-2 and Envisat simulated observations were derived from the NR with a resolution of 7 km between two points along the tracks. An observation white noise of 3 cm RMS was simulated and added to these pseudo observations.

Mercator Ocean operational systems assimilate Sea Level Anomaly (SLA) observations. The Absolute Sea Level (i.e. sea level relative to the geoid) is obtained by using an external Mean Dynamic Topography (MDT) based on the CNES-CLS MDT. In our case, the nature and assimilated runs have different MDTs because of the grid resolution, the model parametrizations and different initialization procedures. We thus chose to assimilate the absolute sea level (witch include the MDT and the SLA) from the NR at 1/12°.

Argo in-situ Temperature and Salinity observations from the surface down to 2000 m were simulated using the 2009 Argo profile positions in the Coriolis CORA3.2 data base (Fig 1b).

**2.4 OSSEs**

The four different OSSEs that have been carried out are summarized in Table 1. The first three simulations address the question of the number of altimeters required to constrain ocean analyses and forecasts. There are three experiments with one
(Jason-2), two (Envisat and Jason-2) and three (Jason-1, Envisat and Jason-2) assimilated satellite data sets. They are respectively called Sat1, Sat2 and Sat3 experiments. The other OSSE addresses the impact of Argo profiling floats together with the three satellite data sets.

All the assimilated experiments start on the 7th of January 2009 and end the 30th of December 2009. The difference between
a given simulation and the NR are used to derive statistics on errors on analyses and forecasts over the last 7 months (June-December 2009). For each assimilation experiment, time series of errors on analyses and forecasts (up to 7 days) are obtained. 7-day forecast errors will be used in this study.

**3. Altimetry OSSEs results**

Impact of assimilation of altimeter data is first analysed on sea level (SL). A wavenumber spectral characterization of the error is also carried out. Errors on surface zonal (U) and meridional velocities (V) are then estimated. Finally, errors on velocities, temperature and salinity at depths are analysed to quantify how the assimilation of multiple altimeter data can constrain deep fields. Analyses are focused on regions with high mesocale variability: Gulf Stream (GS), Agulhas Current (AC) and Kuroshio (KU).

**3.1. Impact on Sea Level**

Figure 2 shows the Mean Square Error (MSE) for the Free Run (FR) and for the analyses and forecasts of the three different assimilation runs (Sat1, Sat2 and Sat3) estimated as the difference with the NR. As expected, the FR shows large differences with the NR as they provide two uncorrelated mesoscale variability fields. Assimilation of one satellite leads to a significant reduction of both analysis and 7-day forecast errors due to a strong correction of the mean sea level. Adding a second
altimeter reduces significantly the errors. The impact of assimilating a third altimeter remains positive but not as large as the addition of a second altimeter. Moreover, errors are largely reduced between the 7-day forecast and the analysis for each of the three assimilation runs.

The evolution in time of the global MSE of Sea Level for both the analysis and 7-day forecast fields is shown on Fig. 3. The system constrained by the 1/12° simulated SSH observations converges toward a stable state in 2 to 3 months. The Free Run

MSE is about 97 cm² (not shown on the plot) over the time period of the experiment; it is reduced to 20 cm² in Sat1. The analysis MSE in Sat2 is lower than Sat1 and approximatively equal to 15 cm². Sat3 provides a slight improvement of a few cm² compared to Sat2. In fact, first altimeter brings the biggest error reduction compared to the Free Run but second and third altimeters keep reducing this error.

To analyse further the structure of errors in high mesoscale variability areas, MSEs for analyses and 7-day forecasts are shown for the GS (Fig. 4 and Fig. 5), AC (Fig. 6 and 7) and KU (Fig. 8 and Fig. 9) regions. "Diamond" like structures can be seen on the analysis error maps for all regions when only one altimeter is assimilated revealing the repetitive spatial sampling of Jason-2. Adding Envisat observations suppresses this effect. In those energetic regions, the MSE for the Free Run is very high in the core of the main current. The increase of the number of assimilated altimeter data set allows a clear
reduction of both 7-day forecast and analysis errors.

To summarize results shown on the different maps, the following score is defined as the MSE for a given AR in percentage of the Free Run MSE:

$$\alpha = 100 \times \left[ \frac{Mean\ Square(error_{AR})}{Mean\ Square(error_{FR})} \right] \tag{1}$$

Those statistics are presented in Table 2.

The greatest impact is made with the assimilation of the first altimeter which strongly reduces the large scale biases existing between the NR and FR. Sat1 Sea Level global analysis MSE reaches 21% of the Free Run MSE. Adding a second satellite (Sat2) reduces the analysis errors by 6%. The third satellite (Sat3) reduces further the errors by about 2%.

Compared to Sat1 global analysis MSE, Sat2 analysis MSE is reduced by 28% and for Sat3 compared to Sat2 error is reduced by 11%. In high eddy energy region, that ratio can reach respectively 42% and 22%.

For a same assimilation experiment, the analysis error is always lower than the 7-day forecast error. The error level of the analysis with one altimeter is close to the 7-day forecast error level when two or three altimeter data sets are assimilated. This is true for all of the considered regions and globally (Table 2). The largest error reduction due to data assimilation occurs in the Agulhas and Kuroshio regions.

The error increase between the analysis and 7-day forecast for each experiment highlights the "model predictability" in the
different regions. The  relative MSE in % between analysis and forecast  increase is 28% globally for Sat1, 35% for Sat2 and 37% for Sat3. In WBCs, values are around 34% for Sat1, around 49% for Sat2 and 54% for Sat3. The error increase is thus the largest when more altimeter data are assimilated.  Analyses are thus better constrained but this does not fully translate into improved forecasts.

Note that as the NR and the ARs use the same atmospheric forcing, 7-day forecast errors are only related to internal
mesoscale dynamics and initialization issues.

## 3.2 Spectral characterization of the error

Estimation of the sea level wavenumber spectrum from altimetry data (e.g. Le Traon et al. 1990; Stammer 1997; Le Traon and Dibarboure, 2008) has allowed major progresses in the characterization of ocean mesoscale dynamics. Wavenumber spectra are used here to characterize sea level analysis and 7-day forecast errors in the Gulf Stream, the Agulhas Current and
the Kuroshio regions.

Wavenumber spectra were calculated from the sea level model error fields using Fast Fourier Transform (FFT). The FFT was applied in 10° x 20° boxes within the previously defined WBCs regions but not fitting exactly to the areas shown on the maps. Longitudinal spectra were estimated from daily error fields and meridionaly averaged. Figure 10 shows the mean sea level error spectrum calculated in the GS(a), AC(b) and KU(c) regions. The computation is made from June to December 2009 both for the analysis and for each 7[th]-day forecast of the assimilation cycle.

The error reduction due to altimeter data assimilation is visible for all of the three selected regions: the free model run error spectrum is higher at all wavelengths larger than 100 km. The assimilation corrects the 1/4° model sea level below its own capacity to represent small scales. Below this limit of 100 km, all the simulations are gathered in one curve. This curve follows the same slope as the full sea level spectrum of the Nature Run (not shown on the plot).

As seen before, the error is reduced each time an additional altimeter is assimilated, for all wavelengths larger than 100 km and up to 1000 km. It is also the case for the analysis compared to the 7-day forecast. Analysis of spectra in a variance preserving form (Figure 11) shows that, compared to analysis errors, 7-day forecast errors occur at larger wavelengths; they have a maximum variance at wavelengths between 300 to 500 km while it is about 200-300 km for analysis errors.

Compared to the Free Run errors, adding one satellite (Sat1) reduces analysis errors for all wavelengths larger than 250 km. Addition of a second (Sat2) and third (Sat3) altimeter allows reducing analysis errors down to 150 km wavelength. In the KU and the GS regions, the Sat2 and Sat3 analysis errors are similar for most of the length scale. In the AG region, the assimilation of the third satellite still allows a significant analysis error reduction.

In most cases, the 7-day forecast error spectrum for the Sat3 experiment is lower than the analysis error for the Sat1 experiment for wavelengths smaller than 300 km.

## 3.3 Impact on surface currents and currents, temperature and salinity at depths

To assess the system ability to reproduce the Nature Run, it is necessary to analyse how non assimilated model variables are improved when assimilating sea level altimeter data. The unobserved variables are impacted by assimilating only sea level observation through two mechanisms. The first one is the multivariate characteristic of the analysis corrections computed by SAM2. The model error covariance matrix is defined with a collection of model anomalies used to calculate increment for all the model prognostic variables, SL, U, V, T and S. The second one is the non linear model dynamics that implies changes on temperature, salinity and velocities when the SSH analysis correction on sea level is added to the model 7-day forecast.

Because of geostrophy, we expect, in particular, that assimilating more altimetry data will better constrain surface velocity fields. Figure 12 presents the MSE of analysis and 7-day forecast for the surface zonal velocity U. The Free Run shows everywhere higher values for the velocity MSEs both for U and V (not shown).

Table 3 shows the same score as the one used for the Sea Level but for the MSE of the analysis and 7-day forecast errors of the zonal and meridional velocity components in $cm^2s^{-2}$. Globally and in the Gulf Stream region, the meridional and zonal velocities MSE are similar, meridional velocity MSE are slightly higher (~10%) than zonal errors in the Agulhas Current and slightly lower (10% again) in the Kuroshio.

The absolute MSEs are decreasing from Sat1 to Sat3, and are much lower than the Free Run. For each experiment, the analysis error is again reduced compared to the 7-day forecast error. The level of error for the 7-day forecast of Sat3 is, in most regions, comparable to the level of the analysis error of Sat1. The assimilation of a second satellite leads to a higher error reduction than the third one, for both analysis and 7-day forecast and in all regions.

Sat1 global analysis velocity MSEs represent 55% of the Free Run MSE. Additional error reductions of 10% and 4% occur for Sat2 and Sat3. In high eddy energy regions (GS, AC, KU), the analysis MSEs are smaller and can reach 35% of the Free Run MSE for Sat1; they continue to be reduced by 13% and 4% for Sat2 and Sat3 (in average in the WBCs).

7-day forecast surface velocity errors are less reduced when an additional altimeter data set is assimilated. They globally represent 64%, 56% and 53% of the Free Run MSE for respectively Sat1, Sat2 and Sat3.

Assimilation of multiple altimeter data does not only improve the surface velocity but also velocity fields at depth. Figure 13 shows global RMS error profiles for U and V. These plots are similar for the two velocity components and show decreasing error profile with depth. There is a clear positive impact of the assimilation of additional altimetry observations up to 2000 meter depth. The improvement brought by each additional satellite is almost uniform on the vertical and even the third altimeter improves the 3D velocity field estimation.

Assimilating sea level altimeter data also improves the temperature and salinity at depths as shown on RMS error profiles for temperature (T) and salinity (S) of Fig. 14. Temperature error profiles show a maximum at the thermocline depth as the salinity error decreases with depth. Globally, Sat1 gives a good improvement for T a depth, comparatively to Free Run with 0.2°C of RMS error in temperature at 200m depth. Sat2 and Sat3 are not distinguishable and only improve the RMS error score by less than 0.05°C. The experiments with altimeter data assimilation only slightly improved Salinity fields. Sea level as measured by altimetry is to a large extent the signature of baroclinic processes and represents an integral of the density anomaly. As density variations are mainly correlated to temperature variations and less salinity variations in most of the ocean regions, this explains why assimilating altimeter data improves the representation of the upper temperature fields (e.g. Guinehut et al., 2012).

**4. OSSE with Argo and altimetry**

Assimilating altimeter data only improves temperature fields and marginally salinity fields but errors remain large. This leads to the next part of the study concerning the Argo1 experiments. This experiment has been designed to answer how a simulated Argo profiles data set allows correcting large scales when they are assimilated with altimetry compared to the Sat3 experiment. Argo floats are designed to monitor large scale and low-frequency variability as described in Roemmich et al. (2009) and the complementarity between remote sensing observation and in situ profiles has been studied in the North Atlantic using OSSEs-like simulations by Guinehut et al. (2004). They showed how well the estimation of 200m T fields was improved thanks to the merging of in situ profiles and altimeter data. Here one wants to assess the global impact of the Argo profiles assimilation using the idealistic configuration of OSSEs in the Mercator Ocean systems. This issue has already been explored using OSEs by Turpin et al (2016). In the latter study, the impact of Argo profiles was assessed using the operational observing array. Three experiments were intercompared, the first one where half of the Argo floats have been removed, the second where all the floats were removed and the last one where all Argo floats were assimilated. The system used in the OSEs (Model combined to an assimilation scheme) is very similar to the one that is used here, meaning 1/4° Nemo model and the SAM2 assimilation scheme. OSEs results showed an increasing improvement in both 7-day forecast and analysis scores when more profiles are assimilated and this mainly in the 0-300m and 700-2000m depth layers.

Profiles in Fig. 15 represent the RMS of the error of T and S for the 7-day forecasts for the global ocean for the OSSEs Argo1 and Sat3. The black line shows the Free Run score. These scores need to be compared with the results of Turpin et al (2016) in Section 3.1.1 and 3.1.2. Profiles shown in the latter uses RMS of innovations meaning, the difference between the observed T and S profiles and the model 7-day forecast values at the observation point over the seven days of the assimilation cycle. This metric can be compared to our 7-day forecast errors, meaning the difference on the ¼° model grid between the 7[th] field of each assimilation cycle with the Nature Run.

It is then expected that scores may differ from one set of experiment to the other. Moreover there are no reasons for the Nature Run to be similar to the ocean state estimated by OSEs and so the results to be exactly the same.

First, Argo profiles go up to 2000 m depth and allow a good large scale constrain of the first 1500 m of the ocean, complementary to altimetry: RMS of the innovation in Argo1 are smaller than in FR and Sat3. The increase of the error at depth in Argo1 shows a weakness of the assimilation scheme that do not find the right correction at depth that will give a good fit to both in situ and altimetry data. Assimilation of a T,S climatology at depth will prevent such errors by adding information on the deep fields that are not or very sparsely observed. S fields are less impacted than T fields because, as we said, density variations are mainly correlated to temperature variations and less salinity variations in most of the ocean regions.

Then, considering these OSE and OSSE results, we see that the given profiles are very similar. As we explained in the previous part, temperature fields at depth are improved compared to the Free Run when altimetric sea level observations are assimilated and this conclusion can also be made when looking at the OSEs results when analysing the corresponding Free Run and RunNa (meaning No Argo) OSEs of Turpin et al. (2016). In the OSSEs, maxima of RMS errors drop from 1.2°C (Free Run) to 0.9°C (Sat3) and in the OSEs, it goes from 1.35°C (Free Run) to 1.18°C (RunNa). For S, both protocols give the similar conclusion that salinity is not highly impacted by altimetry data assimilation.

Improvement brought by the Argo float assimilation is explained by the comparison between Argo1 and Sat3 for OSSEs and the RunOP (for Operationnal Run) and RunNa for Turpin et al. (2016) OSEs. Temperature RMS error maximum reaches 0.6°C for Argo1 and 1°C in the RunOP; in both cases it is reduced compared to simulations without Argo profiles assimilations. Concerning Salinity, maxima are located at the surface and are close to 0.2 psu for Argo1 and 0.17 psu for RunOP. The major improvement is done in Argo1 where the RMS error is divided by almost two compared to Sat3.

This comparison helps to validate the results of the OSSE experiments. The similarity of the error profiles for both the OSE and OSSE is a good indication of the realism of the OSSE experimental context, at least in term of errors relative to the "Nature Run" for the OSSE and the real ocean for OSEs.

Figure 16 maps give a better understanding of how and where the improvements are made in Argo1 compared to Sat3. They represent the RMS error of temperature at the surface, 318m, 902m and 1941m. Those depths correspond to model vertical level. Only fields in the upper 2000m are shown because it is the maximum depth for Argo profiles.

Sat3 RMS error maps show larger scale patterns compared to the Argo1 fields where much more small structures are visible. At the surface, in-situ data assimilation is the most effective in the southern ocean where RMS errors are strongly driven back to a much smaller value (from more than 2°C to less than 0.8°C). Elsewhere Argo1 presents weaker and smaller RMS error compared to Sat3.

The 318m depth is the most impacted level by the assimilation presented here. The strong RMS error in the Atlantic is efficiently corrected in Argo1 and values are reduced everywhere else. Errors show smaller structures and only remain high in the WBCs.

The last two maps (at 902m and 1941m) give similar results but in a much less significant way. Big patterns in Sat3 are corrected and lead to small RMS error structures in Argo1.

Finally we did not comment the impact of Argo observations on the Seal Level since the differences are not significant between Argo1 and Sat3.

## 5. Conclusion

A series of Observing System Simulation Experiments (OSSEs) was carried out with a global data assimilation system at 1/4° resolution using simulated data derived from a 1/12° resolution free simulation. The objective was to quantify how well multiple altimeter missions and Argo can constrain a global data assimilation system. Impact of multiple altimeter data is clearly evidenced. The first altimeter is the one that reduces the most the error and corrects large scale sea level biases. This was also found in OSEs conducted with different real time forecasting systems (e.g. Lea et al., 2014; Oke et al., 2015). ) where the first altimeter contributes the most to the Sea Level error reduction. Forecasts of sea level and ocean currents continue to be improved when moving from one altimeter to two altimeters with a relative error reduction of almost 30%. The addition of a third altimeter still improves the forecasts even at this medium 1/4° resolution and brings an additional relative error reduction of about 10%. Results show that a third altimeter still provides sea level and ocean current error reduction in every highly dynamic area such as WBCs. This is because in WBCs a 1/4° model is not able to create structures with scales smaller than 100-200km, but when assimilating several altimeters, this limit falls closer to 100km. Assimilating altimeter data improves the representation of the upper temperature fields. The addition of Argo has a major impact to improve temperature fields and demonstrates the essential role of Argo together with altimetry to constrain the ocean interior in a global data assimilation system. Salinity fields are only marginally improved. Results derived from these OSSEs are consistent with those derived from experiments with real data (OSEs) but they allow a more detailed analysis of errors. They also show that our OSSEs are well calibrated to simulate the impact of observing systems on our ocean analyses and forecasts.

The study is now being extended to analyse the impact of extension of Argo (deep Argo, improved coverage in western boundary currents and in the tropics), evolution of the altimeter constellation like the use of SAR altimeters with a reduced measurement error compared to the LRM classic observations and impact of the other elements of the global in-situ observing systems (e.g. moorings, gliders).

### Acknowledgements

This study was funded as part of a CNES – Mercator Ocean collaboration. S. Verrier Phd grand was co-funded by Ifremer and Mercator Ocean.

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

35

| Name | Resolution | Assimilation | Data Set | | | |
|---|---|---|---|---|---|---|
| | | | Jason2 | Jason1 | Envisat | Argo |
| Nature Run | 1/12° | Ø | | | | |
| Free Run | 1/4° | Ø | | | | |
| Sat1 | 1/4° | yes | yes | | | |
| Sat2 | 1/4° | yes | yes | yes | | |
| Sat3 | 1/4° | yes | yes | yes | yes | |
| Argo1 | 1/4° | yes | yes | yes | yes | yes |

Table 1 : Computed simulations and assimilated data set.

| SL | GLO | | GS | | AC | | KU | |
|---|---|---|---|---|---|---|---|---|
| | ANA | FCST | ANA | FCST | ANA | FCST | ANA | FCST |
| Sat1 | 21 | 29 | 19 | 29 | 14 | 22 | 13 | 20 |
| Sat2 | 15 | 24 | 12 | 23 | 9 | 18 | 8 | 14 |
| Sat3 | 14 | 21 | 9 | 21 | 7 | 15 | 7 | 14 |

5    Table 2 : Assimilated simulation relative Sea Level MSE in percent of the Free Run MSE.

| U | GLO | | GS | | AC | | KU | |
|------|-----|------|-----|------|-----|------|-----|------|
| | ANA | FCST | ANA | FCST | ANA | FCST | ANA | FCST |
| Sat1 | 53 | 64 | 47 | 62 | 39 | 51 | 35 | 45 |
| Sat2 | 44 | 56 | 34 | 52 | 30 | 44 | 26 | 37 |
| Sat3 | 41 | 53 | 31 | 50 | 27 | 40 | 24 | 36 |
| | | | | | | | | |
| V | GLO | | GS | | AC | | KU | |
| | ANA | FCST | ANA | FCST | ANA | FCST | ANA | FCST |
| Sat1 | 57 | 67 | 57 | 72 | 39 | 48 | 50 | 60 |
| Sat2 | 47 | 59 | 41 | 61 | 30 | 43 | 35 | 48 |
| Sat3 | 42 | 55 | 34 | 56 | 26 | 39 | 32 | 47 |

Table 3 : Assimilated simulation relative zonal velocity (U) meridional velocity (V) and  MSE in percent of the Free Run MSE.

Fig.1 : a) Satellites tracks over 35 days in the North Atlantic. Blue : Jason 2 ; Black : Envisat ; Blue : Jason 1. b) Argo profiles over the year 2009.

Fig.2 : Global Mean Square Error (MSE) of the relative SL in cm$^2$ compared to NR  for the FR (a), Sat1(b,c), Sat2(d,e) and Sat3(f,g). 7-day forecast on the left column and analyses on the right over the  June-December 2009 period

Fig.3 : Time evolution of the global MSE of SL in cm² for both analyses (plain lines) and 7-day forcasts (dashed lines) for Sat1(blue), Sat2(Green) and Sat3(Red).

Fig.4 : GS 7-day forecast MSE of SL in cm² for Sat1(a), Sat2(b) and Sat3(c).

Fig.5 : GS analyses MSE of SL in cm² for Sat1(a), Sat2(b) and Sat3(c).

Fig.6 : AC 7-day forecast MSE of SL in cm² for Sat1(a), Sat2(b) and Sat3(c).

Fig.7 : AC analyses MSE of SL in cm² for Sat1(a), Sat2(b) and Sat3(c).

Fig.8 : KU 7-day forecast MSE of SL in cm² for Sat1(a), Sat2(b) and Sat3(c).

Fig.9 : KU analyses MSE of SL in cm² for Sat1(a), Sat2(b) and Sat3(c).

Fig.10 : Sea Level error energy spectrum in the GS(a), AC(b) and KU(c) for FR(black), Sat1(blue), Sat2(green) and Sat3(red). Analyses are in plain line and 7-day forecast in dashed line.

Fig.11 : Sea Level variance preserving error spectrum in the GS(a), AC(b) and KU(c) for FR(black), Sat1(blue), Sat2(green) and Sat3(red). Analyses are in plain line and 7-day forecast in dashed line.

Fig.12 : Global MSE in cm$^2$.s$^{-2}$ of surface U compared to NR in cm for the FR (a),

Sat1(b,c), Sat2(d,e) and Sat3(f,g). 7-day forecast on the left column and analyses on the right.

Fig.13 : Global 7-day forecast RMSE of U (a) and V(b) profiles in cm.s$^{-1}$ for FR(black), Sat1(blue), Sat2(green) and Sat3(red).

Fig.14 : Global 7-day forecast RMSE of T (a) and S(b) profiles respectively in C° and psu for FR(black), Sat1(blue), Sat2(green) and Sat3(red).

Fig.15 : Global 7-day forecast RMSE of T (a) and S(b) profiles respectively in C° and psu for FR(black), Sat3(blue) and Argo1(green).

Fig.16 : 7-day forecast RMSE of T in °C for Sat3 (left) and Argo1 (right) at the surface (a,b), 318m (c,d), 902m (e,f), and 1941m (g,h).

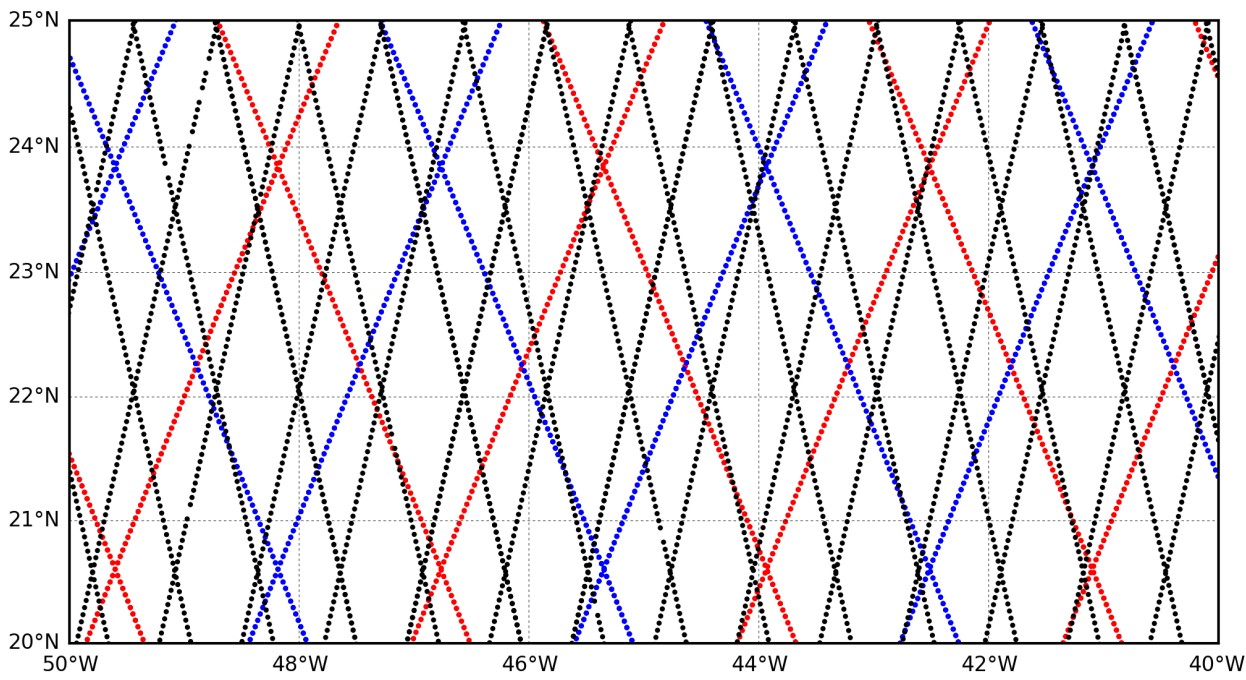

Fig.1 : a) Satellites tracks over 35 days in the North Atlantic. Blue : Jason 2 ; Black : Envisat ; Red : Jason 1.

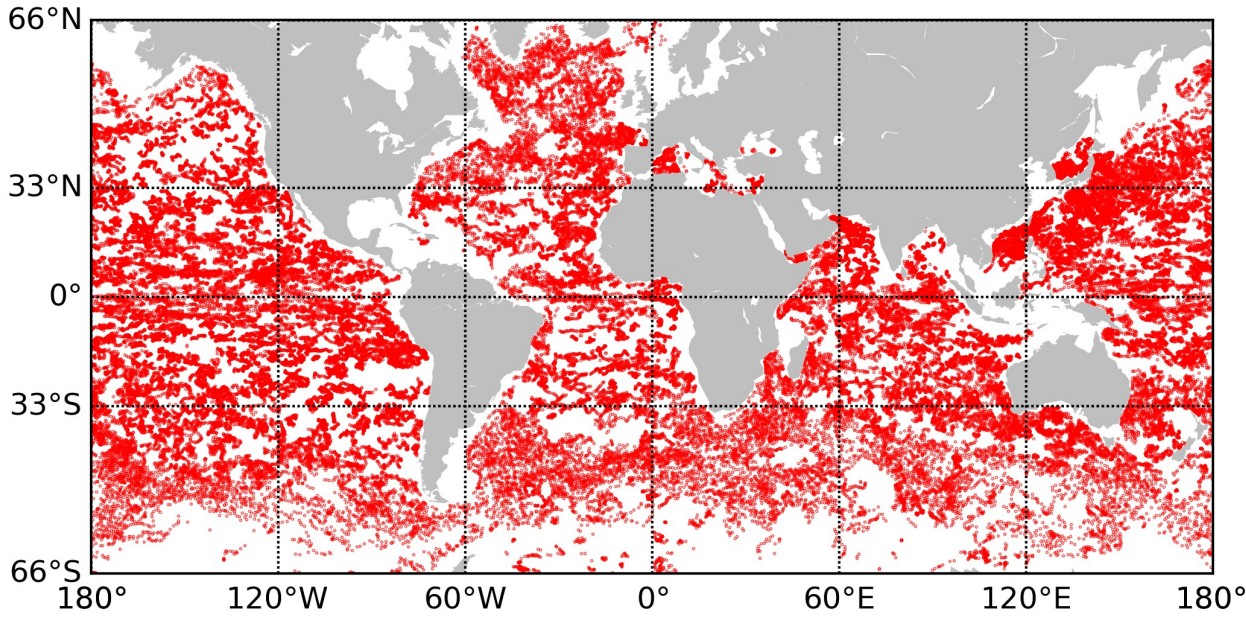

Fig.1 : b) Argo profiles over the year 2009.

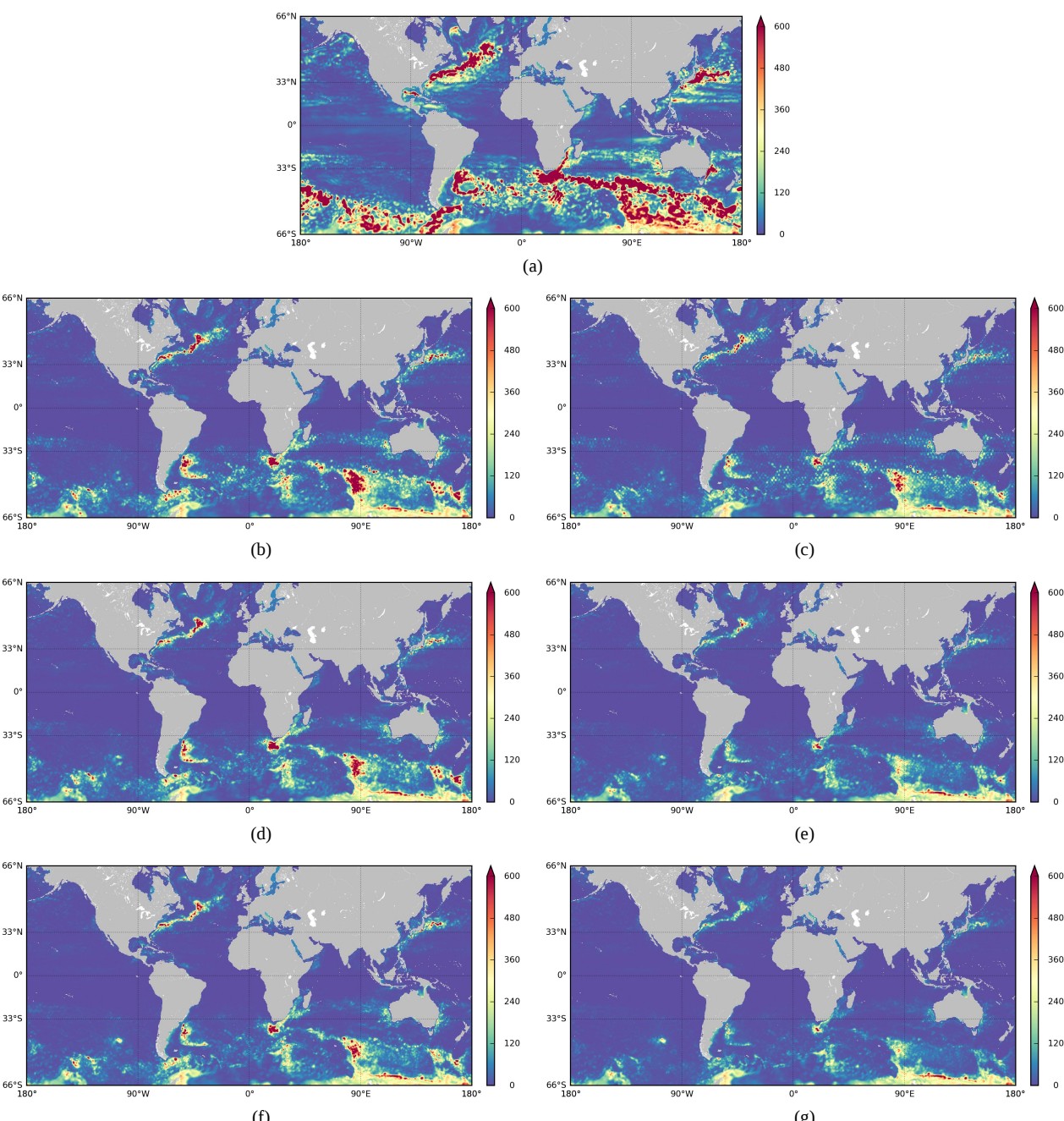

Fig.2 : Global Mean Square Error (MSE) of the relative SL (in cm$^2$) compared to NR for the FR (a), Sat1(b,c), Sat2(d,e) and Sat3(f,g). 7-day forecast on the left column and analyses on the right over the June-December 2009 period.

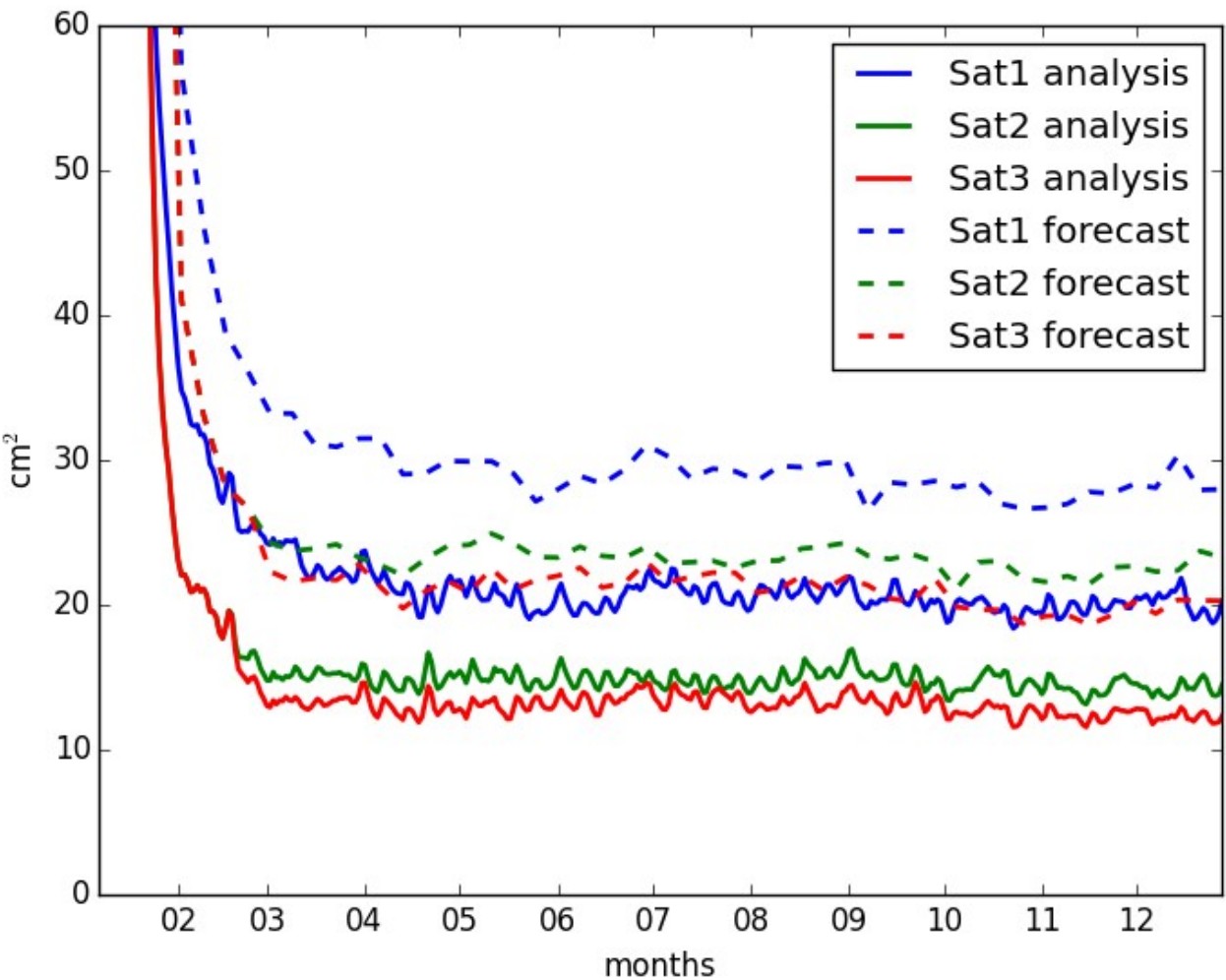

Fig.3 : Time evolution of the global MSE of SL in cm² for both analyses (plain lines) and 7-day forcasts (dashed lines) for Sat1(blue), Sat2(Green) and Sat3(Red).

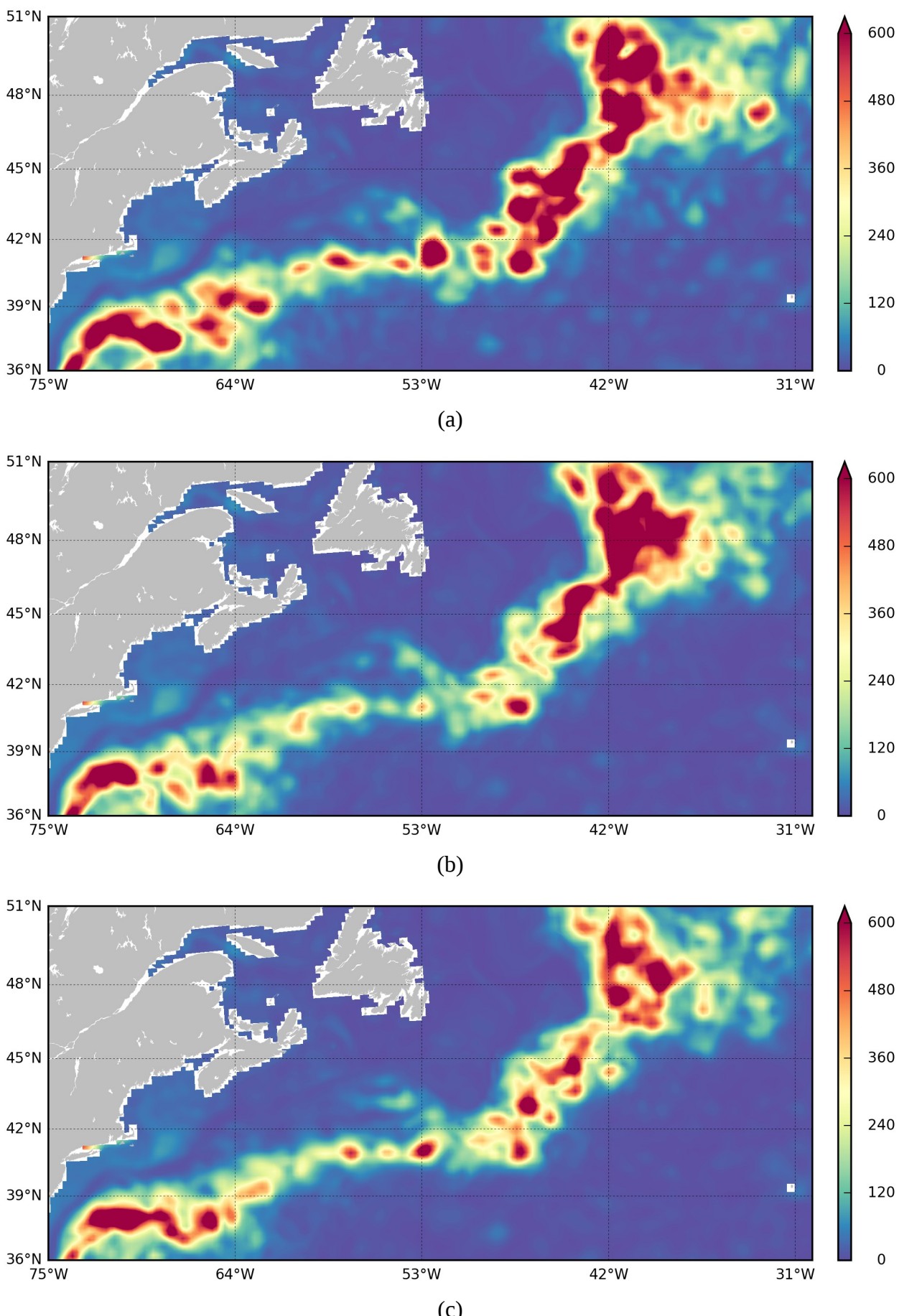

Fig.4 : GS 7-day forecast MSE of SL in cm² for Sat1(a), Sat2(b) and Sat3(c).

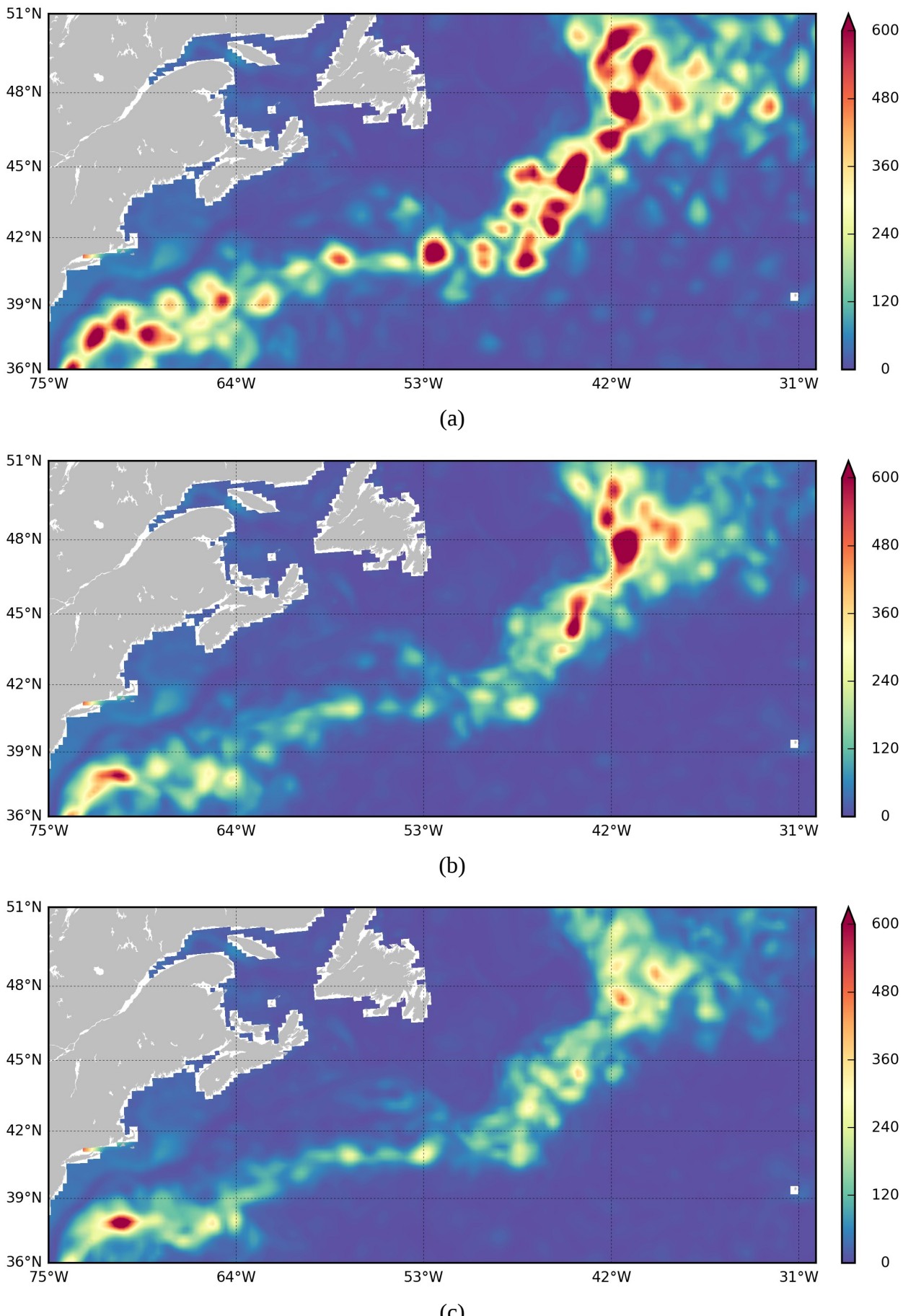

Fig.5 : GS analyses MSE of SL in cm² for Sat1(a), Sat2(b) and Sat3(c).

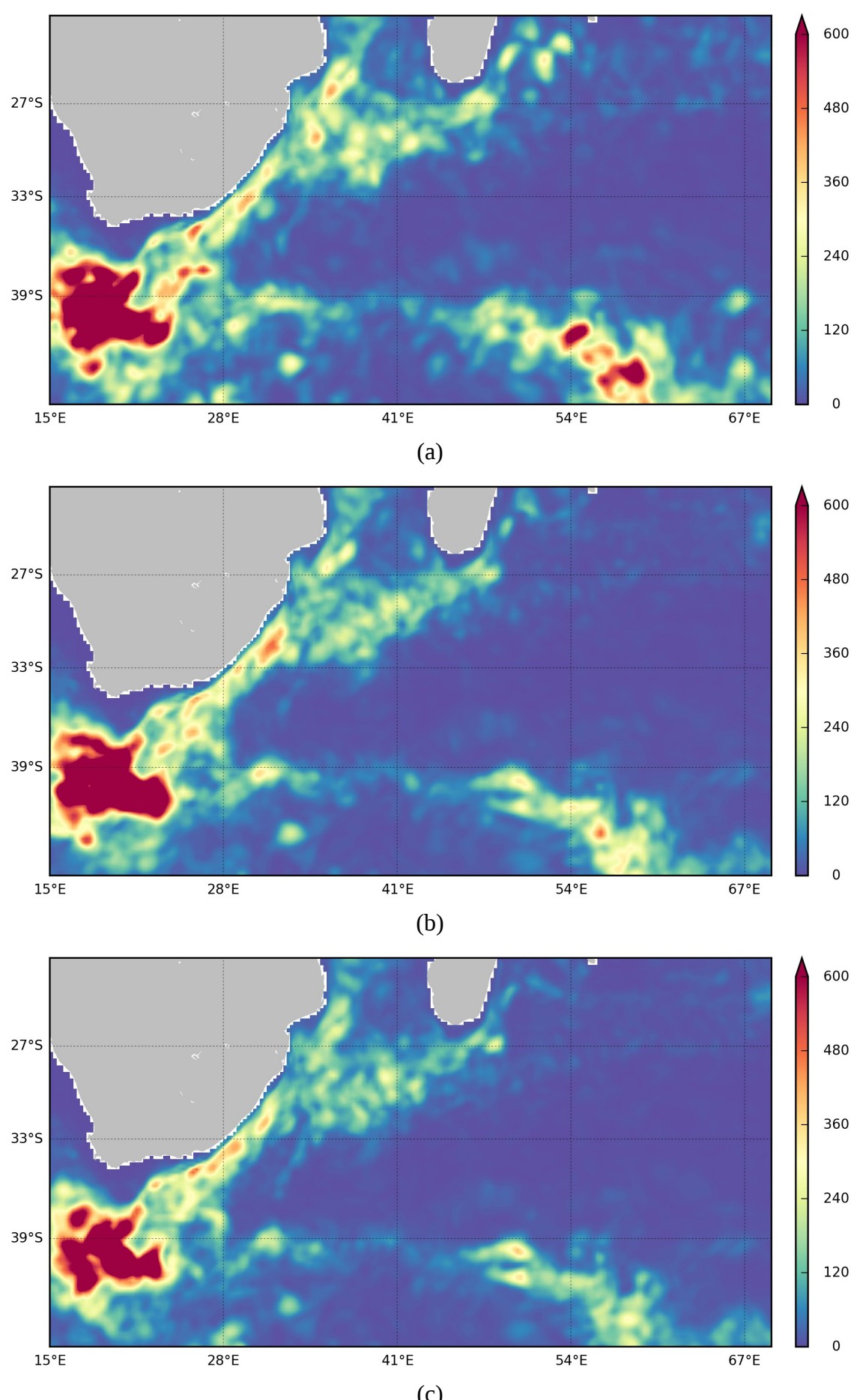

Fig.6 : AC 7-day forecast MSE of SL in cm² for Sat1(a), Sat2(b) and Sat3(c).

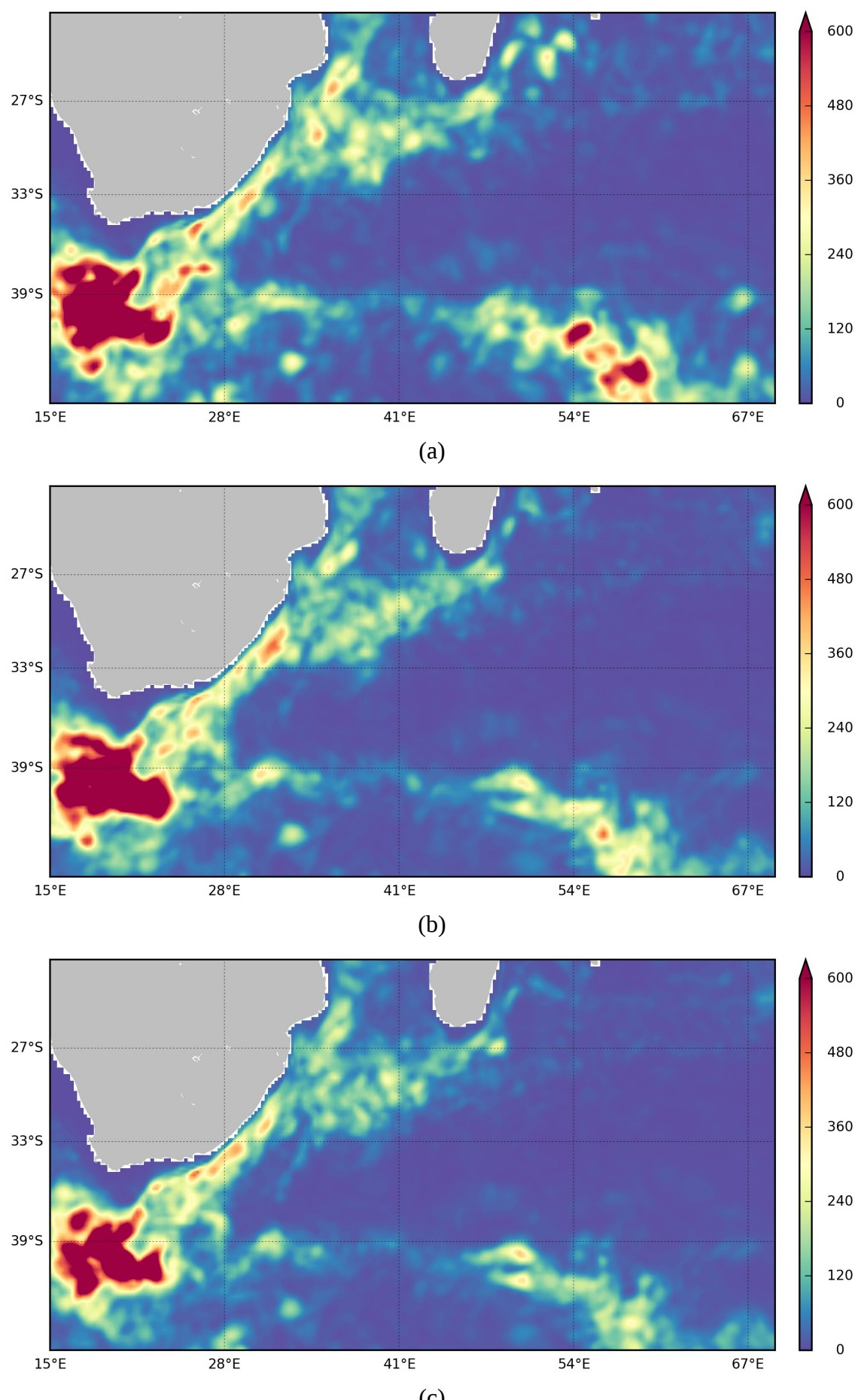

Fig.7 : AC analyses MSE of SL in cm² for Sat1(a), Sat2(b) and Sat3(c).

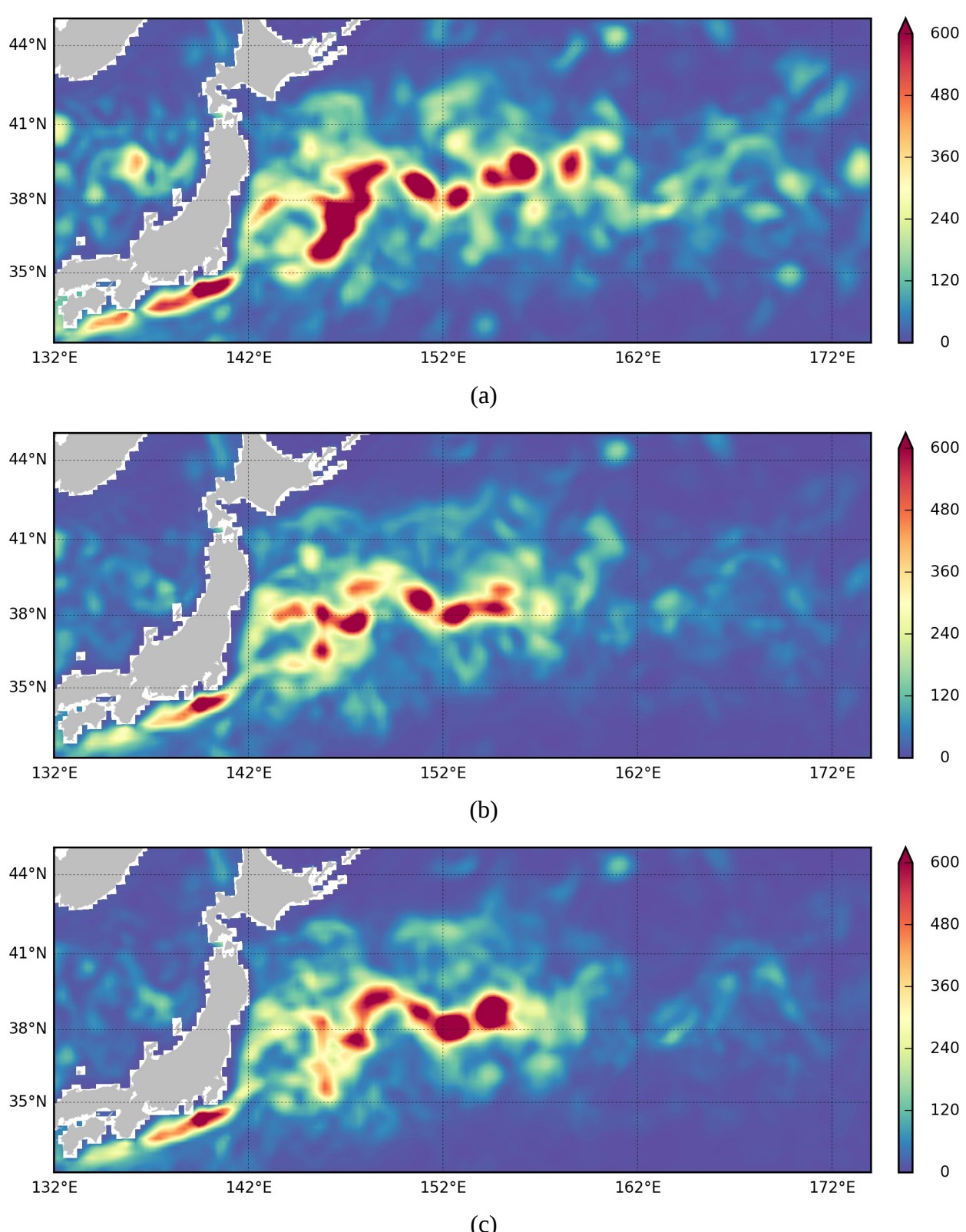

Fig.8 : KU 7-day forecast MSE of SL in cm² for Sat1(a), Sat2(b) and Sat3(c).

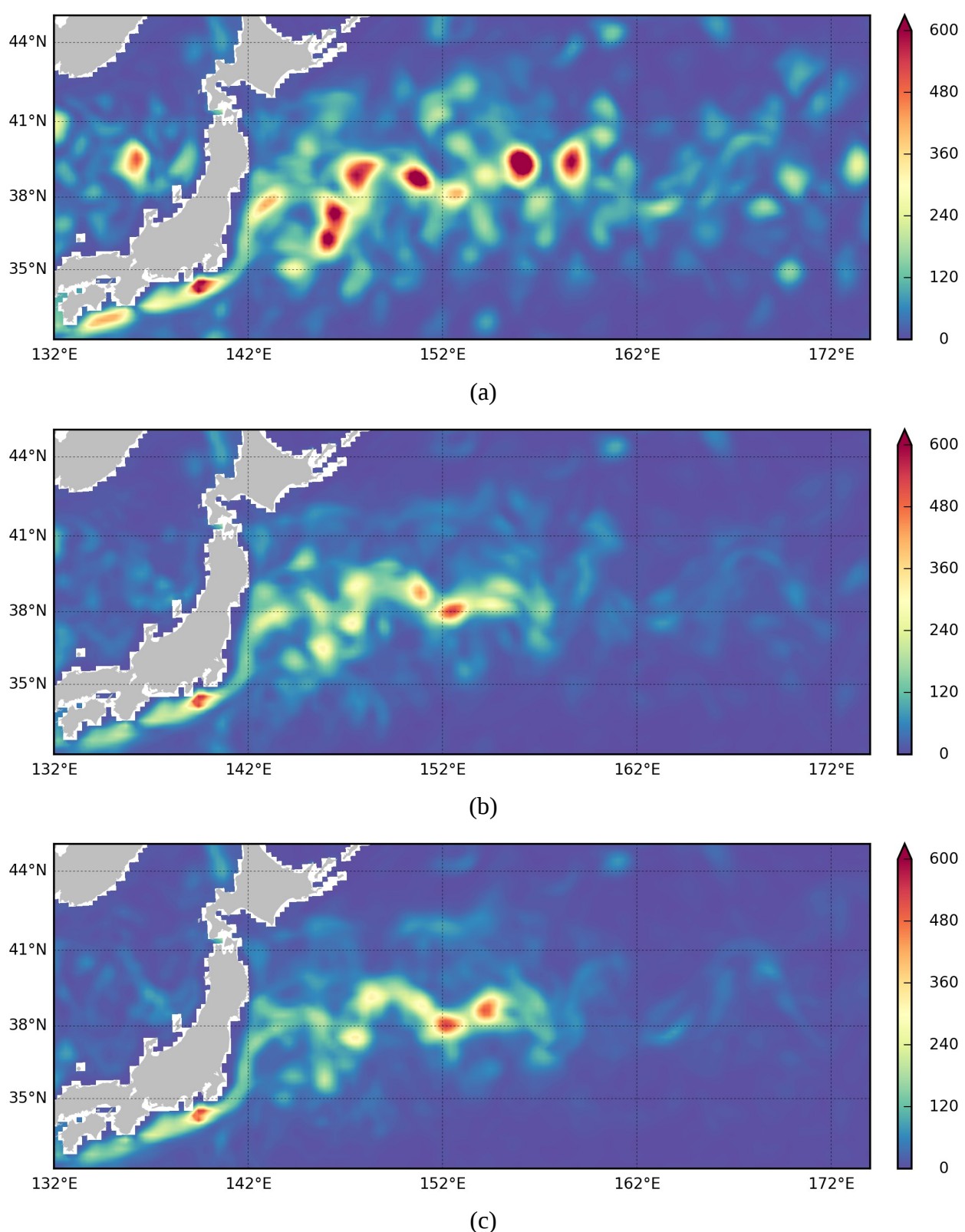

Fig.9 : KU analyses MSE of SL in cm² for Sat1(a), Sat2(b) and Sat3(c).

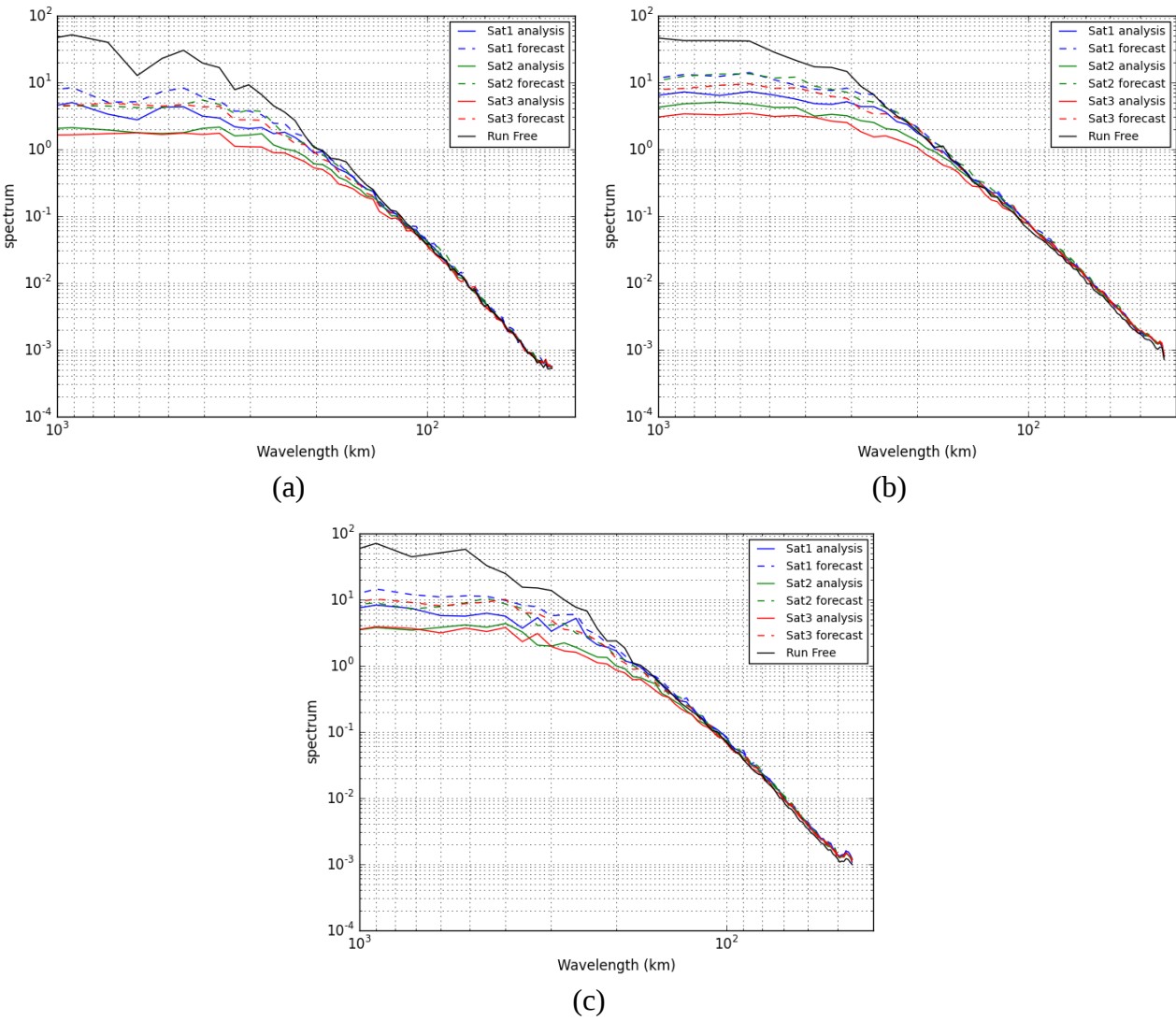

Fig.10 : Sea Level error energy spectrum in the GS(a), AC(b) and KU(c) for FR(black), Sat1(blue), Sat2(green) and Sat3(red). Analyses are in plain line and 7-day forecast in dashed line.

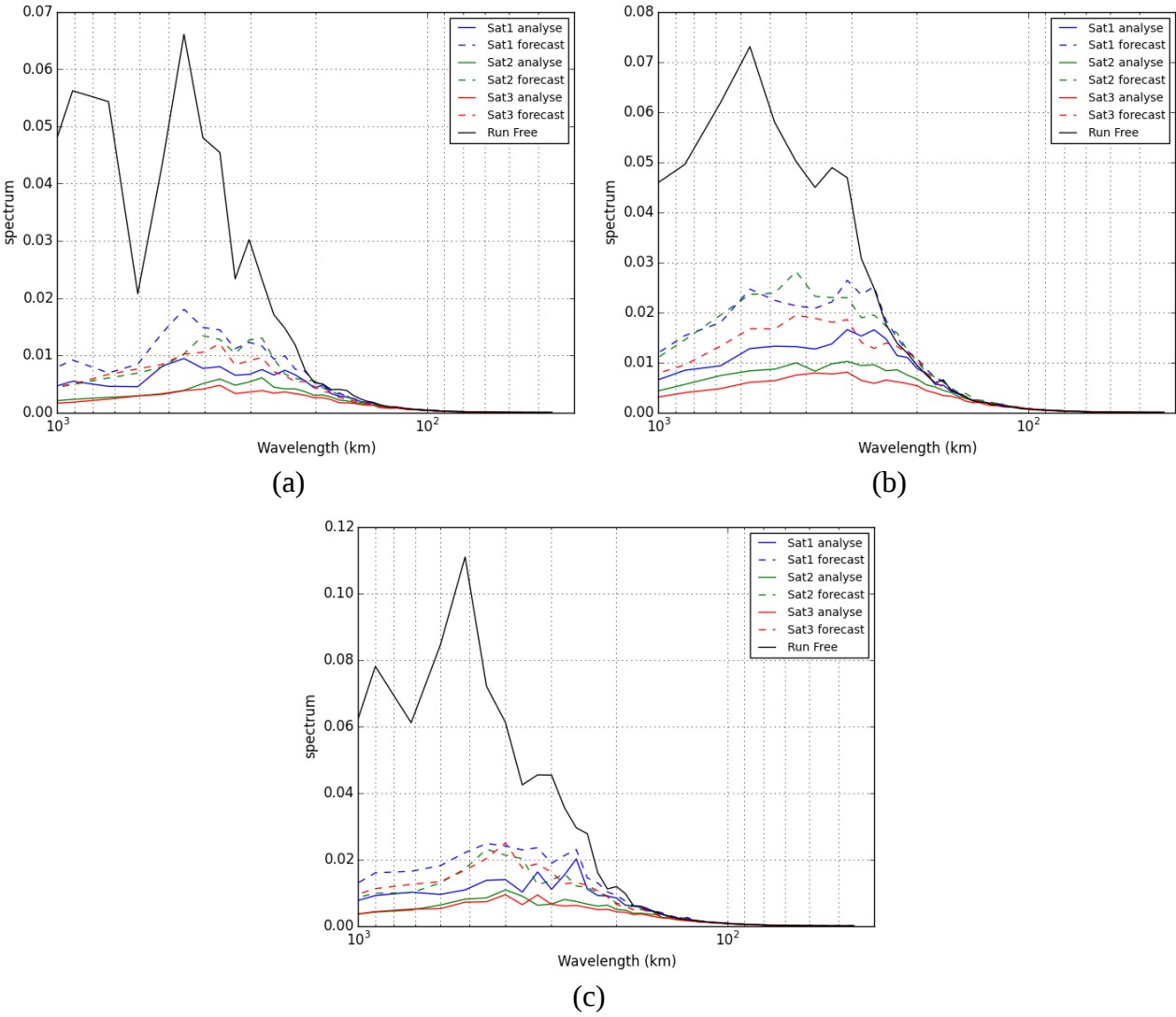

Fig.11 : Sea Level variance preserving error spectrum in the GS(a), AC(b) and KU(c) for FR(black), Sat1(blue), Sat2(green) and Sat3(red). Analyses are in plain line and 7-day forecast in dashed line.

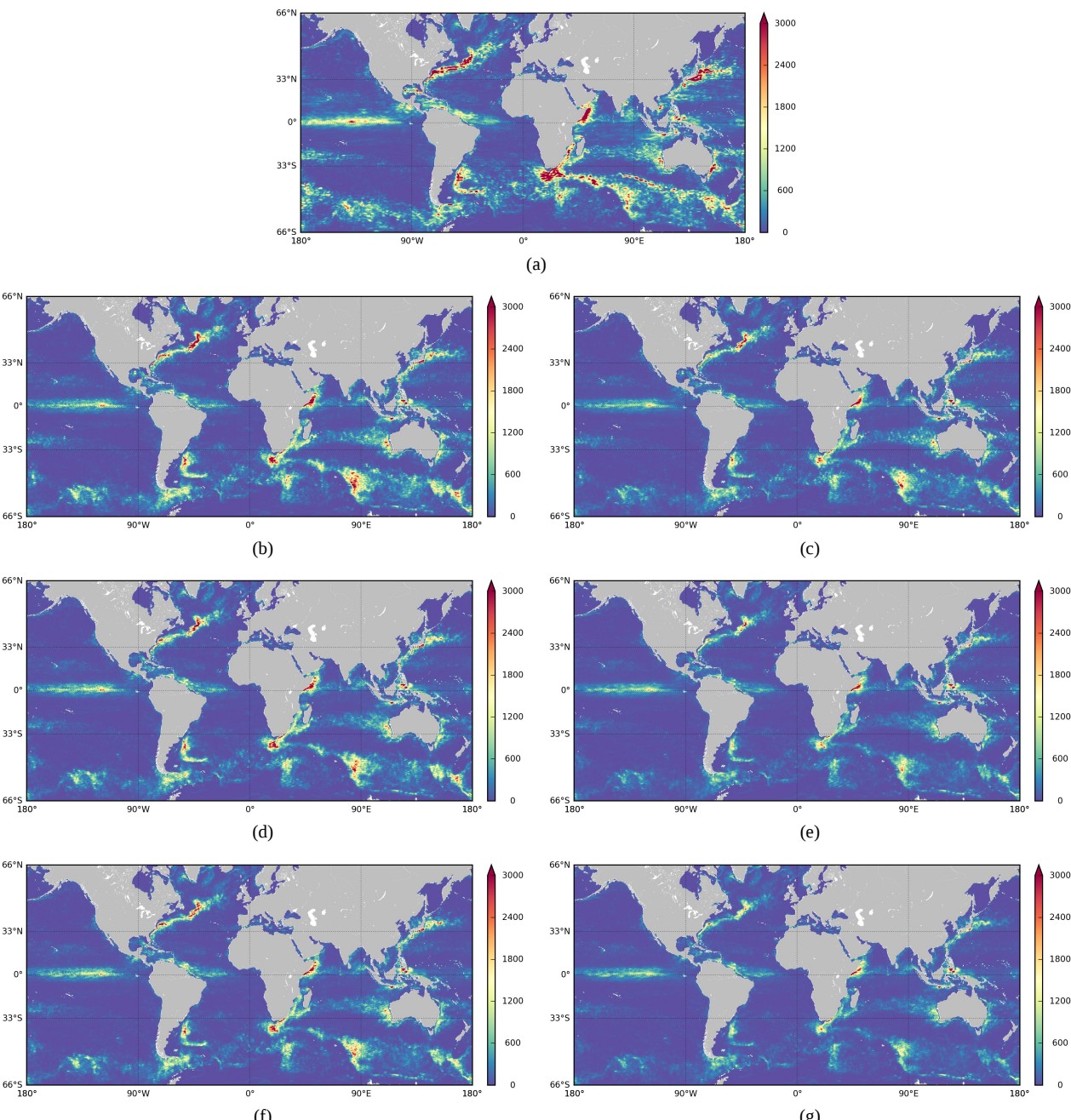

Fig.12 : Global MSE in cm².s⁻² of surface U compared to NR for the FR (a), Sat1(b,c), Sat2(d,e) and Sat3(f,g). 7-day forecast on the left column and analyses on the right.

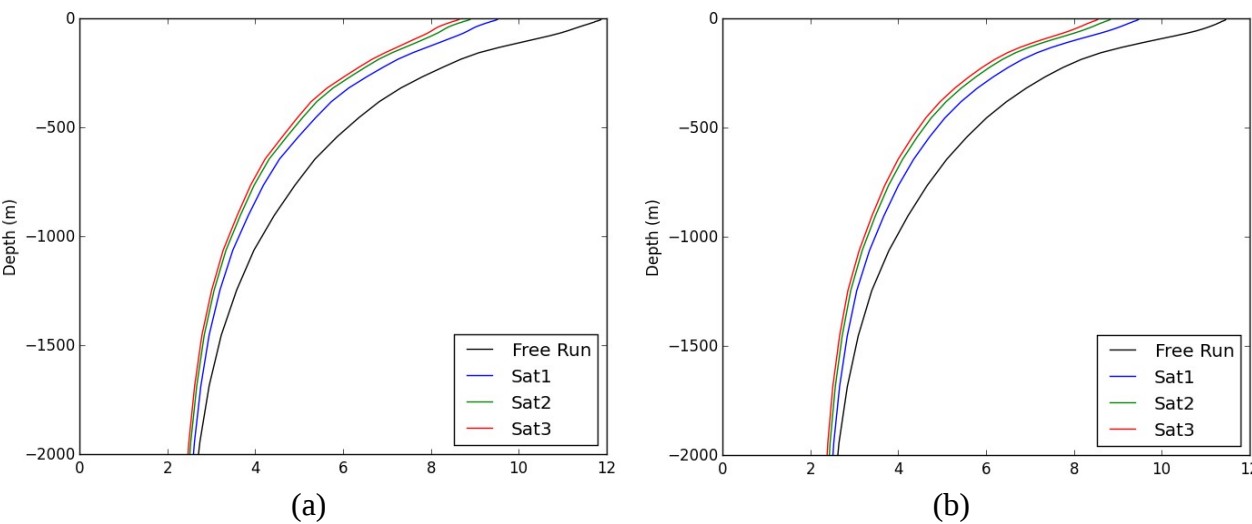

Fig.13 : Global 7-day forecast RMSE of U (a) and V(b) profiles in cm.s$^{-1}$ for FR(black), Sat1(blue), Sat2(green) and Sat3(red).

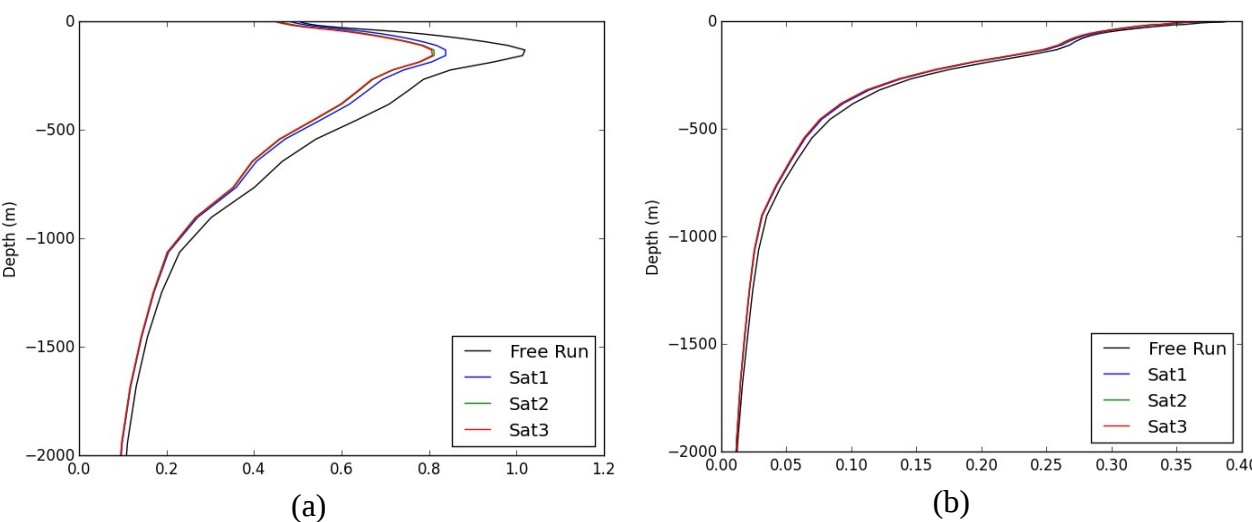

Fig.14 : Global 7-day forecast RMSE of T (a) and S(b) profiles respectively in C° and psu for FR(black), Sat1(blue), Sat2(green) and Sat3(red).

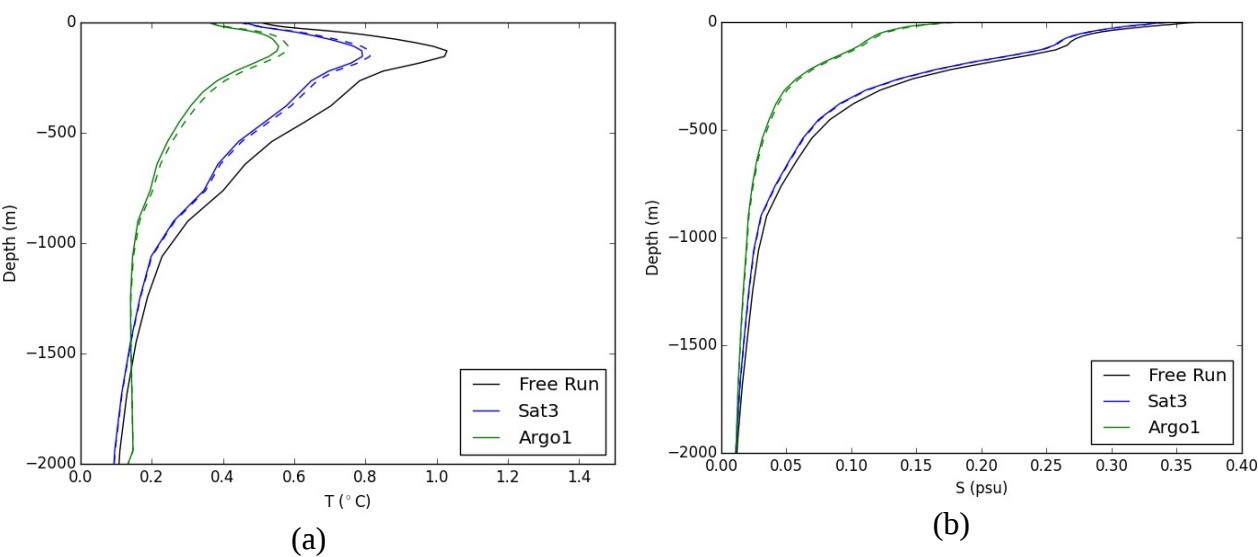

(a)                                                    (b)

Fig.15 : Global 7-day forecast (dashed line) and analysis (plain line) RMSE of T (a) and S(b)
profiles respectively in C° and psu for FR(black), Sat3(blue) and Argo1(green).

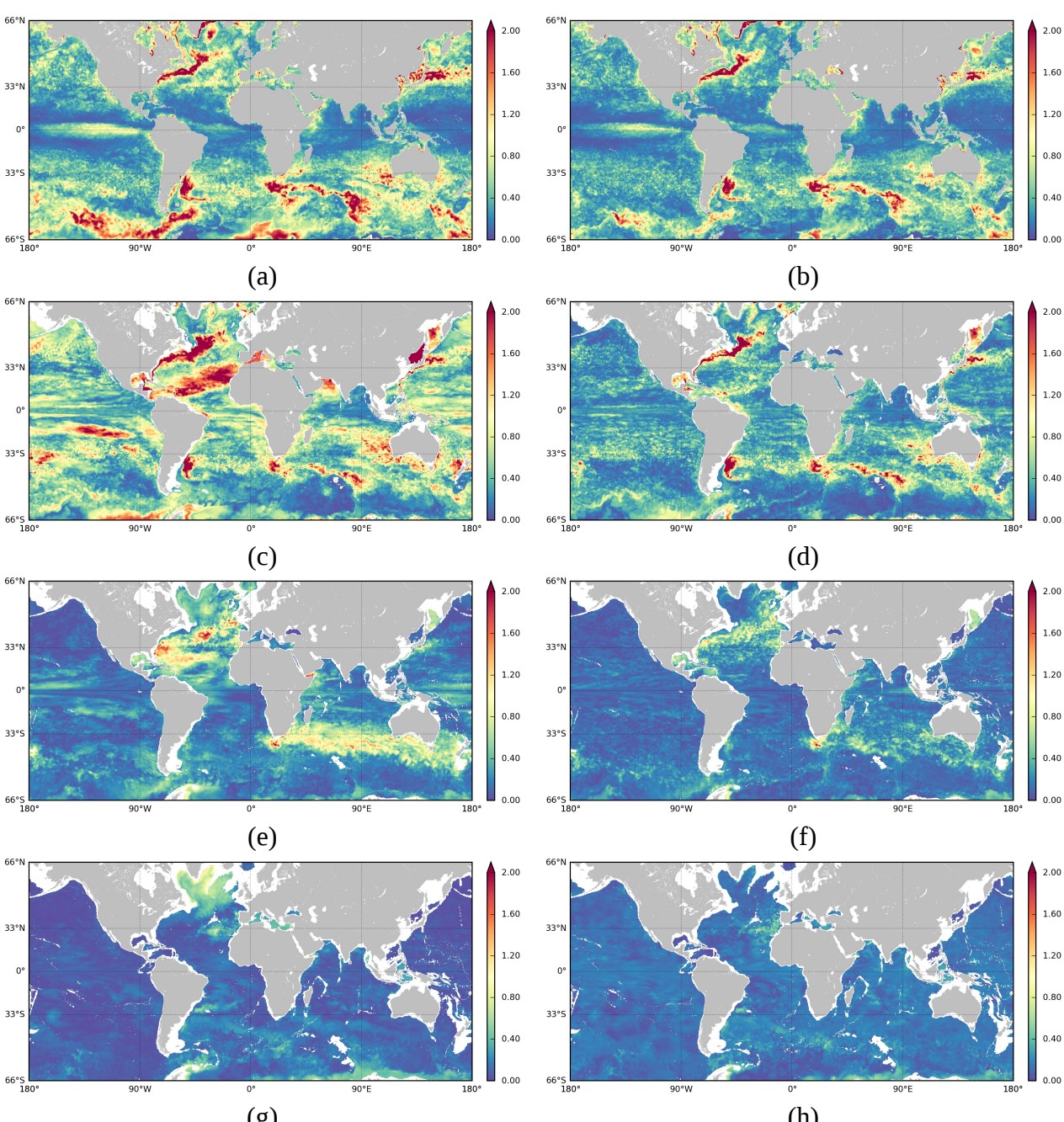

Fig.16 : 7-day forecast RMSE of T in °C for Sat3 (left) and Argo1 (right) at the surface (a,b), 318m (c,d), 902m (e,f), and 1941m (g,h).