# Peer review of "Assessing the impact of multiple altimeter missions and Argo in a global eddy permitting data assimilation system"

_Ocean Science, 2016_

## Referee Comment (RC1) · Anonymous Referee #1 · 12 Mar 2017

Review comments for "Assessing the impact of multiple altimeter missions and Argo in a global eddy permitting data assimilation system" by S. Verrier, P.-Y. Le Traon and E. Remy (os-2016-104).

General Comments

In the paper, Authors addressed how much multiple altimeter data and Argo data constraint a ocean data assimilation system from the simulated observed data of the higher resolution (1/12 degree) free simulation run, which are the observing system simulation experiments (OSSEs). Based on four OSSEs, Authors found that the altimetry data assimilation using simulated Jason 1, 2 and Envisat, increase the accuracy not only of sea surface elevation but also of surface currents, subsurface temperature and salinity

fields. Simulated Argo data in the OSSE, not surprisingly, increase subsurface temperature and salinity fields. OSSE could be an effective tool to diagnosis responses of ocean data assimilation system for considering observations, and this research article would be helpful to understand for the responses. From this review, it seems that this paper would be publishable to Ocean Science after revision with following comments

Specific Comments

O In this paper, the analysis in horizontal distributions are estimated from the differences of each OSSE (Sat1,2,3 and Arg1) from FR (e.g. Fig. 2,4,5,6,7,8,11,15). In order to understand more clearly on the impact of single and multi altimeter date on the assimilation system, it would need the differences among Sat1, 2, 3 as well as the difference from FR.

O page 4, 2nd paragraph from the bottom. It is needed that more explanation on larger improvements of Sat2-Sta1 than one of Sat3-Sat2. This pattern occurs in all comparisons.

O Page 5, 3rd paragraph from top (line 10-11). In the OSSE, this statement is only relevant in the 7-day forecast of this OSSE, and results of the comparison must be changed by the periods of forecast. Is there any more generalized result on this impact?

O Page 5, line 14. "Model predictability" It is needed more detail discussion on "Model predictability". What is the definition of the predictability in this analysis?

O Page 5, 4th paragraph from top (line 14). It is unclear why the MSE increase globally 28% for Sat1, 35% in Sat2, 37% for Sat3. The previous results show that MSE decrease by adding multiple altimeter data. It seems that more discussions on the error are needed.

O page 5, line 28. What is "model surface error fields"? It would be guessable, but in order to understand more clearly, it is needed to clarify what the error field is.

O page 6, 1st paragraph from top. The spectrum of variance preserving form seems to

show different results from the spectrums in Fig.10, especially on impacts in the length scales. The spectrum of variance preserving form would need to be shown, and it need to discuss on the differences of OSSEs in scale between the spectrums.

O Page 7, line 7-8. Model results show the assimilation of altimeter data improved MSEs of subsurface temperature and salinity, and Authors suggest two brief possible mechanisms. The mechanisms would be model and assimilation scheme dependent, so more detail discussion on this mechanism is needed.

O Fig. 4, 5, 6, 7, 8, 9 It seems that more organized figures would be needed rather than separated figures, which is associated with the results in Fig 10.

Technical comments

O Page 2, line 23-24. It seems that this statement is irrelevant.

O page 6, line 22, "MS" would be "MSE"

O Captions of Fig 12, 13, 14 "RF" would be free run or FR
* * *

---

## Referee Comment (RC2) · Anonymous Referee #2 · 29 Mar 2017

The paper described several OSSE experiments where simulated altimeter and Argo profiles are taken from a 1/12 global ocean model and assimilated into a $\frac{1}{4}$ ocean model. 1,2 and 3 altimeter track sets are assimilated and one experiment where all 3 altimeters along with Argo are assimilated based on data from 2009.

This feels a bit like an old study that has not been written up since the question of justifying Multiple altimeters was popular 6-7 years ago. As expected the results improve when assimilating multiple altimeters although this has importantly been shown before assimilating real data in OSE experiments using innovation reductions.

There is no discussion of scientific questions related to these results and I think the study needs such scientific discussion to merit publication so I would say Major revisions. Here are some questions that occurred to me:

Define the scientific problem more rigorously! If you are just asking what's been observed then clearly more altimeters observe more detail. Is that all this paper shows with the sea level errors? To go beyond this you need to discuss how well the model can extrapolate sea level in space and time, either between tracks or from one analysis to the next (forecast). Perhaps there is enough data from 1 altimeter for a good model and good assimilation scheme to extrapolate very well, in which case further altimeters will not lead to improvement? The paper seeks to emphasise the models failure to do this rather than asking how well is has done and whether that's likely to be a limitation for future systems.

Do the 7 day forecasts beat persistence in sea level error for Sat 1, 2,3? The results show that errors grow quickly from the analysis and they grow faster with more altimeters. Sat3 presumably has more small scales which should evolve faster so and be less well captured by persistence? However Page 6 line 4 seems to suggest that the forecast error growth appears strongest at larger scales ∼300km which would run counter to this argument.

The assimilation of absolute sea level page 3 line 38 is likely to have a big impact on the results. Each new altimeter assimilated brings a further "correction" of the MDT towards the 1/12 values. Deserves discussion.

The paper claims to examine complementarity between Argo and altimeters. But the results do not show result for assimilating Argo alone (I suspect Argo 1 results in Fig 14 would be very similar). Are Fig 14 analyses similar or better than forecasts? Also you do not show any impact of Argo on sea level errors? Perhaps forecast errors in sea level might reduce?

Say more about why salinity is not improved? Surely T-S error covariances should give some S improvement if T is improved because a lot of error is heave which preserves T-S relationships?

Minor comments State 7 day forecast times in abstract.

Use of "relative reductions" and MSE reductions rather than using RMSE reductions always relative to the free run makes the % reductions for each satellite appear larger. I think at least the errors in RMSE should also be quoted as this then reflects reductions in sea level units i.e. cm rather than cm^2.

Page 3 line 35 Would altimeter errors not have some error correlation along tracks? Explain more whether the SAM2 assimilation method is still an ensemble method, and how many members are used. This is relevant to capturing T-S correlations for example. You might then explain what "Evolutive" means bottom P2 Also is an analysis and 7 day forecast done every day or are all the diagnostics showing averages of analyses and forecasts at 7 day intervals?

Fig 2 Units should be cm^2. State period in legend Jun-Dec 2009. Text around maps too small to read.

Fig 10 Are these errors MSE or RMSE spectra?

Fig 11 Legend remove "in cm"

Fig 13 coloured lines hard to distinguish. Also would be useful to see this regionally eg GS etc.

Fig 14 Should really include Argo only run

P5 L25 This paragraph should appear earlier in motivation section

P6 L31-35 35%, 13% and 4% all refer to different areas, seeking the largest values. This should be clear in text. Introduce "upto" before each %age for example 65,57,54% appear as 64,56,53% in table

P6 L34 Why are velocity improvements uniform in the vertical?

---

## Referee Comment (RC3) · Anonymous Referee #3 · 10 Apr 2017

Review comments for "Assessing the impact of multiple altimeter missions and Argo in a global eddy permitting data assimilation system" by S. Verrier, P.-Y. Le Traon and E. Remy (os-2016-104).

Based on four OSSEs, Authors assessed the impacts of assimilating multiple altimeters and Argo observations on the global data assimilation system. The simulated observations ( three altimeters and Argo) taken from a high resolution NEMO model were assimilated into low resolution NEMO model. It is not surprised that increasing the altimeter data improved the AR results. Assimilating the derived Argo observations further reduced the bias of the deeper temperature fields relative to the FR. Scientific and presentation qualities should be substantially improved by the major revision of the

manuscript.

Here is some suggestions from me:

Description and explanation of Methods. The authors did not give a clear descriptions of the data assimilation system used in the study. For example, different assimilation window has large impact on the data assimilation results, which is directly related to the observation selections and the disturbed frequency of AR by data assimilation. In this study both observations and anomalies ensemble are different from the Lellouche et al., 2013, why you still used the 7-day time window? Further, Authors used the anomalies constructing method similar to the other studies with EnOI method like Oke et al. (2008). This kinds of method needs large member of samples. How it can save the computation cost compared to the 'EOF' methods in SAM2? How many members have been used? How to select these members? How about the localization? Observation erros covariance? What is the control variables? And so on . . .All these are related to your OSSE results, authors should give a clear descriptions.

Experiment design. The paper discuss both the impact of Sat 1.2.3 and Argo observation systems on data assimilation system. The derived Argo observation effect to the AR results is shown. The corresponding experiment is deigned by assimilating Argo alone or both Argo and satellites? Please clarify it and supply another corresponding experiment. Further, Authors show the three experiments with one (Jason-2), two (Envisat and Jason-2) and three (Jason-1, Envisat and Jason-2) assimilated satellite data sets. The other experiments and analysis with single or combined sat. dataset are also need to be addressed.

Salinity is not improved too much in AR experiments related to the temperature field. The improvements of salinity among three AR experiments (Fig .13) are very small. why? Is it caused by poor T-S background error covariances? The reasons need to be clearly discussed.

The study is mainly focus on the overall impact of assimilating Sat.123 and Argo. The

evolution of impact is also interested for the T,S,U,V in time.

The impact of Argo on the SL also need to be addressed. Perhaps forecast errors in SL might reduce?

Fig3. The Global MSE of SL is fast reduced during the first or two month and then keep small variability. And you explain it "The system constrained by the 1/12° simulated SSH observations converges toward a stable state in 2 to 3 months" Why these happened? Because of the observations coverage or initial conditions or other reasons???

P2 As a following of Turpin et al. (2016), it seems to not true in the beginning of the mansucript : "Analysing the impact of altimetry and Argo in a global data assimilation system through OSSEs has, to our knowledge, not been carried out at least in recent years"

The OSSEs is from January 7, 2009 to end of 2009. So it is not the 1-year OSSEs. Please correct it.

P2 Line '7', "results for existing observing systems must be consistent with those derived from OSSEs.", why must be consistent?

P3 ":::within the upper 100m and with 1m resolution at surface up to 450 m at the bottom:::", make it clearly

P3 Line 20. ".::: our best estimation . . .:::", How about other setup of NEMO or other models, obsevation. Why you say it is the best one. . ..

P5 Line 11 "The error level of the analysis with one altimeter is close to the forecast error level when two or three altimeter data sets are assimilated.". why? One altimeter doesn't work in you AR experiment? Please explain it.

P5 Line 8-9, make it clearly. Why you compare the Sat2 to Sat1 and Sat3 to Sat2, not Sat3 to Sat1?

Please make the unit in Fig2 and other figures same.

---

## Author Comment (AC1) · 31 May 2017

We are thankful for the time and the energy spent to address these comments. The following will answer one by one the comments and questions that have been raised up from the first version of the manuscript.

"In this paper, the analysis in horizontal distributions are estimated from the differences of each OSSE (Sat1,2,3 and Arg1) from NR (?) (e.g. Fig. 2,4,5,6,7,8,11,15). In order to understand more clearly on the impact of single and multi altimeter date on the assimilation system, it would need the differences among Sat1, 2, 3 as well as the difference from FR."

- Our results are based on the comparison between each experiment including the Free Run and the Nature Run (NR). This gives a complete characterization of the errors made by the AR. Figures show differences between the OSSEs and the Nature Run witch represents the truth. We judge that comparing experiments between themselves will make the article too long.

"page 4, 2nd paragraph from the bottom. It is needed that more explanation on larger improvements of Sat2-Sta1 than one of Sat3-Sat2. This pattern occurs in all comparisons."

- The reviewer highlights the fact that improvements are higher when passing from one to two altimeters rather than from two to three altimeters. These scores are relative to error reduction from one experiment to the previous one. In deed first altimeter brings the biggest error reduction compared to the Free Run but second and third altimeters keep reducing this error. We add "In fact, first altimeter brings the biggest error reduction compared to the Free Run but second and third altimeters keep reducing this error" p4 line 35. Those results can also be seen in the description of the error spectrum where the reduction of the error is visible at multiple scales. An explanation is that the first altimeter brings the most important contribution by constraining the large scale mean Sea Surface Height.

"page 5, 3rd paragraph from top (line 10-11). In the OSSE, this statement is only relevant in the 7 day forecast of this OSSE, and results of the comparison must be changed by the periods of forecast. Is there any more generalized result on this impact?"

- Here, the comment deals with the forecast period. As we specify it, we only focus on a 7-day forecast window but we will point it out again in the manuscript. We do not have more general results concerning the period of forecast except that error growth is close to a linear evolution in function of the time.

"page 5, line 14. "Model predictability" It is needed more detail discussion on "Model predictability". What is the definition of the predictability in this analysis?"

- This comment is about the definition beneath the ÂńÂǎmodel predictabilityÂǎÂż. Here it highlights the forecasting capability of the forecasting system. We replaced the sentence in the manuscript by "The error increase between the analysis and forecast for each experiment highlights the model forecasting capabilities at 7 days in the different regions"

"page 5, 4th paragraph from top (line 14). It is unclear why the MSE increase globally 28% for Sat1, 35% in Sat2, 37% for Sat3. The previous results show that MSE decrease by adding multiple altimeter data. It seems that more discussions on the error are needed."

- We need to rephrase our sentence. In deed it was not clear enough. When we write ÂńÂǎMSEÂǎÂż, it is in fact "relative MSE in % between analysis and forecast and not errors between experiments" p5 l19. "page 5, line 28. What is "model surface error fields"? It would be guessable, but in order to understand more clearly, it is needed to clarify what the error field is."

- Concerning this comment, we complete our sentence by specifying that error fields refer to the difference between AR and Nature Run. We corrected it as follow: ÂńÂǎWavenumber spectra were calculated from the sea level error fields (AR - NR)ÂǎÂż.

"page 6, 1st paragraph from top. The spectrum of variance preserving form seems to show different results from the spectrums in Fig.10, especially on impacts in the length scales. The spectrum of variance preserving form would need to be shown, and it need to discuss on the differences of OSSEs in scale between the spectrums."

- As suggested by the reviewer, we add the variance preserving spectrum to complete the previous spectrum. Conclusions are not changed but the figures show more clearly that "The error reduction due to altimeter data assimilation is visible for all of the three selected regions: the free model run error spectrum is higher at all wavelengths larger than 100 km."

"page 7, line 7-8. Model results show the assimilation of altimeter data improved MSEs of subsurface temperature and salinity, and Authors suggest two brief possible mechanisms. The mechanisms would be model and assimilation scheme dependent, so more detail discussion on this mechanism is needed."

- This comment is about how well sub surface temperature is improved compared to salinity by assimilating altimetry observations. Except in the Gulf Stream, salinity is not significantly improved when assimilating altimetry data (cf supplementary figures joined to this answer). It is because in the system, sea level errors are well correlated to upper temperature errors and less to salinity's through the model covariance error matrix. We modified the end of the paragraph like this: "As density variations are mainly correlated to temperature variations and less salinity variations in most of the ocean regions, this explains why assimilating altimeter data improves the representation of the temperature fields (e.g. Guinehut et al., 2012)."

"fig. 4, 5, 6, 7, 8, 9 It seems that more organized figures would be needed rather than separated figures, which is associated with the results in Fig 10."

- Here it is more a suggestion is about re-organizing multiple maps together. We found that this way of representation allows a better visibility of the impact of the assimilation on the mesoscale in each region.

"page 2, line 23-24. It seems that this statement is irrelevant."

- Finally we are not sure to understand what is irrelevant in our statement. We write about OSSEs and not OSEs, that have been done in fact extensively done in the past but not OSSEs.

- About the technical comment, we modified and corrected what has been pointed out.

[Figure]

Salinity RMSE (Experiment – NR) at 318 m in psu

Salinity RMSE (Experiment – NR) profiles in Gulf Stream

**Fig. 1.**

---

## Author Comment (AC2) · 31 May 2017

The reviewer made some comments about our study and the associated manuscript. We are thankful for the time and the energy spent to address these comments. The following will answer one by one the comments and questions that have been raised up from the first version of the manuscript.

"Define the scientific problem more rigorously! If you are just asking what's been observed then clearly more altimeters observe more detail. Is that all this paper shows with the sea level errors? To go beyond this you need to discuss how well the model can extrapolate sea level in space and time, either between tracks or from one analysis to the next (forecast). Perhaps there is enough data from 1 altimeter for a good model

and good assimilation scheme to extrapolate very well, in which case further altimeters will not lead to improvement? The paper seeks to emphasize the models failure to do this rather than asking how well is has done and whether that's likely to be a limitation for future systems."

- Our study is in deed revisiting works that was carried out 10 years ago. We are using here assimilation systems that are very close to systems that are running operationally. As explained in introduction this allow us to much better characterize analysis and forecast errors. This can only be done through OSSEs. Since 10 years, the forecasting system has deeply changed and conclusions have changed in consequence (concerning small scales for examples), these changes needed to be assess. Also we analyzed both velocity, temperature and salinity errors analysis and forecast. One of the objective is to better understand how the data assimilation system work. We modify the abstract to better emphasis the goal of the study.

"Do the 7 day forecasts beat persistence in sea level error for Sat 1, 2,3? The results show that errors grow quickly from the analysis and they grow faster with more altimeters. Sat3 presumably has more small scales which should evolve faster so and be less well captured by persistence? However Page 6 line 4 seems to suggest that the forecast error growth appears strongest at larger scales âĹij300km which would run counter to this argument."

- This point deals with forecast and analysis improvements when adding a new altimeter. We did compute FS relative to persistence and in general, except in WBCs, FS beats persistence. This probably reflects that at this resolution ($1/4°$) the model has not enough skill to forecast small scale features. Looking at the variance preserving error spectrum (new fig in the manuscript and Sup1 in the supplementary figures joined to this answer), it appears that the error growth between analysis and forecast occurs around scales of 300km. We do not have explanation on the forecast error that grows faster with more altimetry data assimilated.

"The assimilation of absolute sea level page 3 line 38 is likely to have a big impact on the results. Each new altimeter assimilated brings a further "correction" of the MDT towards the 1/12 values. Deserves discussion."

- The comment is about assimilating the MDT. Our results do not account for MDT errors and this should be taken into account when comparing our results with those derived from operational system (in OSEs). Analyzing the effect of MDT error (that are quite complex to characterize) on ocean analysis and forecasting is a topic by it self. But we found that the first altimeter allows the largest improvement, then the additional altimeters do not bring further improvement on the MDT. We modified the end of the paragraph by : "We thus chose to assimilate the absolute sea level (witch include the MDT and the SLA) from the NR at 1/12°".

"The paper claims to examine complementarity between Argo and altimeters. But the results do not show result for assimilating Argo alone (I suspect Argo 1 results in Fig14 would be very similar). Are Fig 14 analyses similar or better than forecasts? Also you do not show any impact of Argo on sea level errors? Perhaps forecast errors in sea level might reduce?"

- The reviewer recommends making a new simulation using only the Argo float and it is true we did not run an OSSE with Argo data alone. Analyses and forecasts are now compared in a new Figure 14. The Argo simulation is made to assess the complementarity concerning deep fields of in situ observations assimilated in addition of altimetry data. We rephrase the sentence p7 l18 to highlight that. Sea level results are not significantly changed by the assimilation of Argo observations.

"Say more about why salinity is not improved? Surely T-S error covariances should give some S improvement if T is improved because a lot of error is heave which preserves T-S relationships?"

- This comment is about how well sub surface temperature is improved by assimilating altimetry observations compared to salinity. Except in the Gulf Stream (Sup2 and Sup3

in the supplementary figures joined to this answer), salinity is not significantly improved when assimilating altimetry data. It is because in the system, sea level errors are well correlated to upper temperature errors and less to salinity's through the model covariance error matrix. Those covariances are build from a free model simulation. We modified the end of the second paragraph p7 like this: "As density variations are mainly correlated to temperature variations and less salinity variations in most of the ocean regions, this explains why assimilating altimeter data improves the representation of the temperature fields (e.g. Guinehut et al., 2012)."

"Use of "relative reductions" and MSE reductions rather than using RMSE reductions always relative to the free run makes the % reductions for each satellite appear larger. I think at least the errors in RMSE should also be quoted as this then reflects reductions in sea level units i.e. cm rather than cmËE̦2."

- The reviewer suggests using RMSE but our results in % are given from MSE, same score using RMSE would have been different. For assimilation system, scores are always given in MSE. We agree that error reduction would have been smaller when considering RMSE. Also the altimeter community is more naturally considering energy rather than the RMS.

"page 3 line 35 Would altimeter errors not have some error correlation along tracks? Explain more whether the SAM2 assimilation method is still an ensemble method, and how many members are used. This is relevant to capturing T-S correlations for example. You might then explain what "Evolutive" means bottom P2 Also is an analysis and 7 day forecast done every day or are all the diagnostics showing averages of analyses and forecasts at 7 day intervals?"

- Finally, we did not simulate correlated errors along tracks that Is why we used a white noise. We will better explain the assimilation system used in our experiments. Here we have used 349 members in a fixed pre-computed basis (ensemble of model anomalies) and their selection is explained in Lellouche et al. 2013. We add the explanation of

"Evolutive" in the text (model error covariance propagated by the dynamical model). Analyses results are given for the 7 days of the cycle and forecast (one forecast is made each 7 days) results are given only for th 7th (last) day of the cycle.

Other comments deal with lay out suggestions or small corrections that we made in the manuscript.

——————————————————

[Figure]

[Figure]

Sup 1 : Variance preserving error spectrums in Gulf Stream (GS), Aghulas (Ag) and Kuroshio (Ku)

[Figure]

Sup 2 : Salinity RMSE (Experiment - NR) at 318 m in psu

Sup 3 : Salinity RMSE (Experiment - NR) profiles in Gulf Stream in psu

**Fig. 1.**

[Figure]

---

## Author Comment (AC3) · 31 May 2017

The reviewer made some comments about our study and the associated manuscript. We are thankful for the time and the energy spent to address these comments. The following will answer one by one the comments and questions that have been raised up from the first version of the manuscript.

"Description and explanation of Methods. The authors did not give a clear descriptions of the data assimilation system used in the study. For example, different assimilation window has large impact on the data assimilation results, which is directly related to the observation selections and the disturbed frequency of AR by data assimilation. In this study both observations and anomalies ensemble are different from the Lellouche et

al., 2013, why you still used the 7-day time window? Further, Authors used the anomalies constructing method similar to the other studies with EnOI method like Oke et al. (2008). This kinds of method needs large member of samples. How it can save the computation cost compared to the 'EOF' methods in SAM2? How many members have been used? How to select these members? How about the localization? Observation errors covariance? What is the control variables? And so on... All these are related to your OSSE results, authors should give a clear descriptions."

- Concerning the assimilation scheme we used, the SAM2 description will be filled out in the manuscript. We kept the setup of the assimilation scheme as it is in the operational system and described in Lellouche et al. 2013 except for the representativity errors we did not take into account here, the assimilation of the full SSH signal and not only the SLA and the uniform observing error covariance matrix (3 cm in RMS).Control variables are the Sea Level, Zonal and Meridional speeds, Temperature and Salinity.Yes, the method is closed to the EnOI used by Oke et al. (2008). SAM2 does not used EOF but a fixed basis of model anomalies is pre-computed. It saves calculation time compared to a classical evolutive filter method. We have used 349 members in a fixed pre-computed basis and their selection and localization are explained in Lellouche et al. 2013. Our filter is not evolutive in the way that error is not propagated by the model. The anomaly basis changed at each analysis cycle : they follow the global model climatology. Analyses results are given for the 7 days of the cycles and forecasts results are given only for th 7th (last) day of the cycle (one forecast is made each 7 days).

"Experiment design. The paper discuss both the impact of Sat 1.2.3 and Argo observation systems on data assimilation system. The derived Argo observation effect to the AR results is shown. The corresponding experiment is deigned by assimilating Argo alone or both Argo and satellites? Please clarify it and supply another corresponding experiment. Further, Authors show the three experiments with one (Jason-2), two (Envisat and Jason-2) and three (Jason-1, Envisat and Jason-2) assimilated satellite data

sets. The other experiments and analysis with single or combined sat. dataset are also need to be addressed."

- This comment deals with experiments design. We chose to not compute a simulation assimilating only Argo since the subject of the study deals with increasing altimetry observations and the complementarity of Argo observations with altimetry, we made that clearer in the introduction. Moreover, experiment design is specified in Table 1. We only computed the experiment assimilating Argo observations and the 3 altimeters data as we found that changing the number of satellites did not change much the T,S error profiles (fig. 13).

"Salinity is not improved too much in AR experiments related to the temperature field. The improvements of salinity among three AR experiments (Fig .13) are very small. Why? Is it caused by poor T-S background error covariance ?The reasons need to be clearly discussed."

- This comment is about how well sub surface temperature is improved by assimilating altimetry observations compared to salinity. Except in the Gulf Stream, salinity is not significantly improved when assimilating altimetry data. It is because in the system, sea level errors are well correlated to upper temperature errors and less to salinity's through the model covariance error matrix. We modified the end of the second paragraph p7 like this: "As density variations are mainly correlated to temperature variations and less salinity variations in most of the ocean regions, this explains why assimilating altimeter data improves more the representation of the temperature fields (e.g. Guinehut et al., 2012)." T/S relationship is in fact embedded in the background error covariance matrix built from a free model long simulation. Though T and S variations are linked to the density through the density equation used in the ocean model (here NEMO, using the UNESCO density equation). Temperature changes have a much larger effect on density than salinity changes in most of ocean regions. The SSH is changed by density changes through the steric effect.

"The study is mainly focus on the overall impact of assimilating Sat.123 and Argo. The evolution of impact is also interested for the T,S,U,V in time."

- This suggestion is about time evolution of the error. Because we did not want to put too many figures, we only selected profiles and not time evolutions. In both point of view, results lead to the same conclusions. Error on observed variables decreases during the first 6 months (on average) and then keep a constant level. Non observed variables errors are gradually reduce with time but slower than the seal level errors, in many places those errors do not reach a constant value after the almost one year of assimilation.

"The impact of Argo on the SL also need to be addressed. Perhaps forecast errors in SL might reduce?"

- The reviewer suggests to assess the impact of Argo observation on SL scores but we did not show it because it is not significant. We add this information in the manuscript (Last lines of the 4th part).

"fig3. The Global MSE of SL is fast reduced during the first or two month and then keep small variability. And you explain it "The system constrained by the 1/12 âŮe simulated SSH observations converges toward a stable state in 2 to 3 months" Why these happened? Because of the observations coverage or initial conditions or other reasons???"

- The reviewer ask for more explanation concerning the error reduction in time as it can be seen in the figure 3. As long as the observing errors is a fixed at 3cm, only the differences between the initial states at each cycle of the OSSEs and the NR explains the convergence. The fact that we assimilate more observations make this convergence stronger.

"p2 As a following of Turpin et al. (2016), it seems to not true in the beginning of the manuscript : "Analysing the impact of altimetry and Argo in a global data assimilation

system through OSSEs has, to our knowledge, not been carried out at least in recent years""

- Here we address the problem with OSSEs and not OSEs that have been done extensively in recent years.

"The OSSEs is from January 7, 2009 to end of 2009. So it is not the 1-year OSSEs. Please correct it."

- For this point we corrected it in "almost one year".

"p2 Line 7, "results for existing observing systems must be consistent with those derived from OSSEs.", why must be consistent?"

- This comment address the fact that if OSSEs and OSEs do not show the same results it implies that OSSEs are not correctly calibrated. Here the goal is to study how a system close to the operational one works in an ideal case. Thus the errors of our OSSEs need to be close to the error of the OSEs when assimilating real observations.

"p3 "...within the upper 100m and with 1m resolution at surface up to 450 m at the bottom...", make it clearly"

- The reviewer suggests to clarify the vertical size of the grid. m The vertical resolution increases from 1 m for the surface layer to 450 m at 5000 m depth. We changed it in the manuscript.

"p3 Line 20. "... our best estimation ...", How about other setup of NEMO or other models, observation. Why you say it is the best one..."

- The NEMO at 1/12° of resolution is at the state of the art in term of high resolution oceanconfiguration. Comparison with other high resolution model such as HYCOM were conducted and shows globally the same level of quality. Those simulations are very realistic. In fact the free model estimations are not the best for the surface and subsurface ocean variability. The best are the analysis that includes both model and

observation information. So we remove the word "best" and replace it by "good".

"p5 Line 11 "The error level of the analysis with one altimeter is close to the forecast error level when two or three altimeter data sets are assimilated.". why? One altimeter doesn't work in you AR experiment? Please explain it."

- This comment is misunderstanding that if analysis error with one satellite is close to the forecast error with 2 satellites do not mean that the one satellite simulation does not "work. For each data assimilation experiment, the analysis error is lower than the forecast error showing the benefit of the data assimilation. It happens that the forecast error level with two assimilated altimeters is close to the analysis error level with one altimeter. We do not have explanation on the reason why.

"p.5 Line 8-9, make it clearly. Why you compare the Sat2 to Sat1 and Sat3 to Sat2, not Sat3 to Sat1?"

- We do not compare Sat3 to Sat1 because we assess the improvements brought by each new altimeter and do not want to add more figures.

And we finally changed the unit in Fig2.

---

## Referee Report (RR1)

The manuscript has been improved in some aspects. However, besides of possible further stylistic cosmetics, there is still a couple of important obscurities left, which need to be clarified.

The authors have added some contents in the description of methods. For example, the number of ensemble samples and the control variables. There is a explanation of the SAM in Lellouche et al., 2013. And Lellouche et al( 2013) has denoted the details of producing the model anomaly fields. However, to easy following in this manuscript, the authors still need to give a clear description of setup in these OSSEs. For example, the selection of the model anomaly fields from which period ( which year to which year)? It is good to know how "The anomaly basis changed at each analysis cycle : they follow the global model climatology." ? Further, is the SAM2 system the same as that SAM in Lellouche et al., 2013? if not ,what is the difference? If the same, why here it called SAM2? The SAM has different version, however, it will be good to use a consistent name.

The authors chose to not compute a simulation assimilating only Argo. The main object of this study is to "Assessing the impact of multiple altimeter missions and Argo in a global eddy permitting data assimilation system". So both Argo and altimeter observations have been used in the analysis in Argo1. The authors have compared the this study and OSE study and the difference between Sat3 and Argo1 and denoted the contribute of Argo observations to AR. However, the corresponding experiment is deigned by assimilating Argo alone and altimeter observations alone is easy and clear to understand how the Argo contribute to the improvement in the experiment Argo1.

At line 24-26 in Page 3, "The 1/12° free model is chosen for NR ::: in the ocean compared to a 1/4° resolution simulation." Is there reference can be used to prove it?

"The explain of less change of the salinity in AR relative to temperature", Does it is also related to the NEMO model itself? There is also the similar case in other region using the NEMO model, whiling the other model (like HYCOM) has stronger correlation of S with the change of sea surface variables like (Sea-level,SST).

The evolution of impact of AR is still interested for the T,S,U,V in time. Please add corresponding contents in manuscript or in the response to the referee.

in page 4, How to pre-calculate Mean Dynamic Topography? Please explain it.

in Fig15, the RMSE of the Argo1 temperatures are changed from smaller in upper layers to larger in deep layers relative to the FR. And this case doesn't appear at the performance of salinity. What is the reason?

Further, in Fig 16, the RMSE of the 7-day forecast of T in 1941m is obvious smaller in Argo1 compared to that in Sat3. Why is there different performance in the same experiment between Fig15 and Fig 16 in deep layers?

The language improvement needs to be addressed, For example:
At line 16 in page 4 , "All the assimilated experiments start ON the 7th of January 2009 and end ON the 30th of December 2009"

---

## Author Response (AR2)

The manuscript has been improved in some aspects. However, besides of possible further stylistic cosmetics, there is still a couple of important obscurities left, which need to be clarified.

- The authors have added some contents in the description of methods. For example, the number of ensemble samples and the control variables. There is a explanation of the SAM in Lellouche et al., 2013. And Lellouche et al( 2013) has denoted the details of producing the model anomaly fields. However, to easy following in this manuscript, the authors still need to give a clear description of setup in these OSSEs. For example, the selection of the model anomaly fields from which period ( which year to which year)? It is good to know how "The anomaly basis changed at each analysis cycle : they follow the global model climatology." ?

> Anomalies are calculated from a 10 year free model and at each date they are equal to the difference between the free run and a running mean. At the date of an analysis,  only anomalies within  the past 30 days and future 30 days and from the different years are considered.  The final number of anomalies that are kept for given analysis is equal to 349. This means that the anomaly basis changes at each analysis date and follows the global model climatology.
These anomalies are selected accordingly (not respectively) to the season of the assimilation cycle (in a way) to get a basis (statistics) evolving consistently with the climatology. Our filter is not evolutive as the model error covariance IS NOT propagated by the dynamical model. The model correction is calculated as a linear combination of the selected anomalies.

- Further, is the SAM2 system the same as that SAM in Lellouche et al., 2013? if not ,what is the difference? If the same, why here it called SAM2? The SAM has different version, however, it will be good to use a consistent name.

> SAM2 refers to SAM « version » 2 which is the same SAM version described in the Lellouche et al. paper

- The authors chose to not compute a simulation assimilating only Argo. The main object of this study is to "Assessing the impact of multiple altimeter missions and Argo in a global eddy permitting data assimilation system". So both Argo and altimeter observations have been used in the analysis in Argo1. The authors have compared the this study and OSE study and the difference between Sat3 and Argo1 and denoted the contribute of Argo observations to AR. However, the corresponding experiment is deigned by assimilating Argo alone and altimeter observations alone is easy and clear to understand how the Argo contribute to the improvement in the experiment Argo1.

> The article emphasizes the constrain bring by an altimetry constellation. The first three experiments (Sat1 to Sat3) focus on the impact of the number of altimeters assimilated and the 3D ocean reconstruction without any other data. The last experiment Argo1 was designed to see how the additional constrain bring by in situ Argo T/S profiles will improve the analysis.
The error statistics of the OSE study consisting of assimilating real altimetry and in situ data is compared the OSSE Argo1 which also assimilates altimetry and in situ data. The underlying idea is the check the realism of the OSSE setup by comparing the level of errors in both experiments when the same kind of data is assimilated.  Adding an « Argo alone OSSE » was not the purpose of this article and would add further analysis.

- At line 24-26 in Page 3, "The 1/12° free model is chosen for NR ::: in the ocean compared to a 1/4° resolution simulation." Is there reference can be used to prove it?

Hulburt et al 2009 compared ORCA12 and ORCA025 simulations

- "The explain of less change of the salinity in AR relative to temperature", Does it is also related to the NEMO model itself? There is also the similar case in other region using the NEMO model, whiling the other model (like HYCOM) has stronger correlation of S with the change of sea surface variables like (Sea-level,SST).

> All OGCM that have a similar equation of state (EOS or TEOS) will have similar correlation between the temperature, salinity and density. The correlation patterns will depend on the ocena region.

- The evolution of impact of AR is still interested for the T,S,U,V in time. Please add corresponding contents in manuscript or in the response to the referee.

> We did not judge that adding more figures such as time evolution of mean and RMS may add to the analysis. Even though you will find at the end of this document some of them leading to equivalent conclusions to the article.

- in page 4, How to pre-calculate Mean Dynamic Topography? Please explain it.

> « pre-calculated Mean Dynamic Topography (MDT). » : change le texte comme suit :
external Mean Dynamic Topography (MDT) based on the CNES-CLS MDT

- in Fig15, the RMSE of the Argo1 temperatures are changed from smaller in upper layers to larger in deep layers relative to the FR. And this case doesn't appear at the performance of salinity. What is the reason?

> Argo profiles go up to 2000 m depth and allow a good large scale constrain of the first 1500 m of the ocean, complementary to altimetry: RMS of the innovation in Argo1 are smaller than in FR and Sat3. The increase of the error at depth in Argo1 shows a weakness of the assimilation scheme that do not find the right correction at depth that will give a good fit to both in situ and altimetry data. Assimilation of a T,S climatology at depth will prevent such errors by adding information on the deep fields that are not or very sparsely observed. S fields are less impacted than T fields because Sea level as measured by altimetry is to a large extent the signature of baroclinic processes and represents an integral of the density anomaly. As density variations are mainly correlated to temperature variations and less salinity variations in most of the ocean regions, this explains why assimilating altimeter data improves the representation of the upper temperature fields"

- Further, in Fig 16, the RMSE of the 7-day forecast of T in 1941m is obvious smaller in Argo1 compared to that in Sat3. Why is there different performance in the same experiment between Fig15 and Fig 16 in deep layers?

> Both figures are coherent : at 1941m the RMS of the temperature innovation for Argo1 is larger than for the Free Run (Figure 15) and Sat3 (Figure 15 and 16).

[Figure]

Global analysis RMS of salinity error (psu) around the 100m depth level from June to December
Free Run Sat1 Sat2 Sat3

[Figure]

Global analysis RMS of temperature error (C°) around the 100m depth level from June to December
Free Run Sat1 Sat2 Sat3

[Figure]

Global analysis RMS of zonal velocity error (m.s$^{-1}$) around the 100m depth level from June to December
Free Run Sat1 Sat2 Sat3

Time evolution of some analysis RMS error for the global ocean. Results lead to same conclusion as spatial sight.

[revised manuscript text omitted]